# A remarkable adaptive paradigm of heart performance and protection emerges in response to marked cardiac-specific overexpression of ADCY8

Kirill V Tarasov[1], Khalid Chakir[1], Daniel R Riordon[1], Alexey E Lyashkov[2], Ismayil Ahmet[1], Maria Grazia Perino[1], Allwin Jennifa Silvester[1], Jing Zhang[1], Mingyi Wang[1], Yevgeniya O Lukyanenko[2], Jia-Hua Qu[1], Miguel Calvo-Rubio Barrera[2], Magdalena Juhaszova[1], Yelena S Tarasova[1], Bruce Ziman[1], Richard Telljohann[1], Vikas Kumar[1], Mark Ranek[3], John Lammons[1], Rostislav Bychkov[1], Rafael de Cabo[2], Seungho Jun[3], Gizem Keceli[3], Ashish Gupta[3], Dongmei Yang[1], Miguel A Aon[1], Luigi Adamo[3], Christopher H Morrell[1], Walter Otu[1], Cameron Carroll[1], Shane Chambers[1], Nazareno Paolocci[3], Thanh Huynh[4], Karel Pacak[4], Robert Weiss[3], Loren Field[5], Steven J Sollott[1], Edward G Lakatta[1]*

[1]Laboratory of Cardiovascular Science, National Institute on Aging, National Institutes of Health, Baltimore, United States; [2]Translational Gerontology Branch, Intramural Research Program, National Institute on Aging, National Institutes of Health, Baltimore, United States; [3]Division of Cardiology, Department of Medicine, Johns Hopkins University School of Medicine, Baltimore, United States; [4]Section on Medical Neuroendocrinology, Eunice Kennedy Shriver National Institute of Child Health and Human Development, National Institutes of Health, Bethesda, United States; [5]Kraennert Institute of Cardiology, Indiana University School of Medicine, Idianapolis, United States

*For correspondence: lakattae@grc.nia.nih.gov

**Competing interest:** The authors declare that no competing interests exist.

**Abstract** Adult (3 month) mice with cardiac-specific overexpression of adenylyl cyclase (AC) type VIII (TG[AC8]) adapt to an increased cAMP-induced cardiac workload (~30% increases in heart rate, ejection fraction and cardiac output) for up to a year without signs of heart failure or excessive mortality. Here, we show classical cardiac hypertrophy markers were absent in TG[AC8], and that total left ventricular (LV) mass was not increased: a reduced LV cavity volume in TG[AC8] was encased by thicker LV walls harboring an increased number of small cardiac myocytes, and a network of small interstitial proliferative non-cardiac myocytes compared to wild type (WT) littermates; Protein synthesis, proteosome activity, and autophagy were enhanced in TG[AC8] vs WT, and Nrf-2, Hsp90α, and ACC2 protein levels were increased. Despite increased energy demands in vivo LV ATP and phosphocreatine levels in TG[AC8] did not differ from WT. Unbiased omics analyses identified more than 2,000 transcripts and proteins, comprising a broad array of biological processes across multiple cellular compartments, which differed by genotype; compared to WT, in TG[AC8] there was a shift from fatty acid oxidation to aerobic glycolysis in the context of increased utilization of the pentose phosphate shunt and nucleotide synthesis. Thus, marked overexpression of AC8 engages complex, coordinate adaptation "circuity" that has evolved in mammalian cells to defend against stress that threatens health or life (elements of which have already been shown to be central to cardiac ischemic pre-conditioning and exercise endurance cardiac conditioning) that may be of biological significance to allow for proper healing in disease states such as infarction or failure of the heart.

## Editor's evaluation

The study is overall well-planned and the amount of data presented by the authors is impressive. The work nicely incorporates animal-level physiology (echocardiography data), tests for known canonical markers of hypertrophy, and then delves into an unbiased analysis of the transcriptome and proteome of LV tissue in bulk. The techniques and analyses in the study are adequately executed and within the realm of expertise of the Lakatta laboratory. This study is a necessary and crucial first step to extensively phenotype this mouse line and generate hypotheses for further work.

## Introduction

Adaptations have evolved in all organisms to cope with both acute and chronic internal and environmental stress. For example, during acute exercise increased autonomic sympathetic-mediated AC/cAMP/PKA/Ca$^{2+}$ signaling, the quintessential mediator of both acute and chronic stress, also activates acute physiologic adaptations to moderate the exercise-induced increase in sympathetic signaling. In response to repeated bouts of acute exercise induced AC/cAMP/PKA/Ca$^{2+}$ stress, chronic adaptations emerge (endurance exercise conditioning) (*Vega et al., 2017*; *Venturaclapier et al., 2007*; *Lovallo, 2005*).

Prolonged and intense chronic cAMP-mediated stress in experimental animal models results in cardiomyopathy and death (*Zhang et al., 2013*; *Antos et al., 2001*). During chronic pathophysiologic states for example chronic heart failure (CHF) in humans, AC/cAMP/PKA/Ca$^{2+}$ signaling progressively increases as the degree of heart failure progresses, leading to cardiac inflammation, mediated in part, by cyclic-AMP- induced up-regulation of renin-angiotensin system (RAS) signaling. Standard therapies for CHF include β-adrenoreceptor blockers and RAS inhibitors, (*Squire and Barnett, 2000*; *Werner and Böhm, 2008*) which although effective, are suboptimal in amelioration of heart failure progression.

One strategy to devise novel and better therapies for heart failure, would be to uncover the full spectrum of concentric cardio- protective adaptations that becomes activated in response to severe, chronic AC/cAMP/PKA/Ca$^{2+}$-induced cardiac stress. The young adult TG$^{AC8}$ heart in which cardiac-specific over expression in mice of Adenylyl Cyclase (AC) Type 8, driven by the α myosin heavy chain promoter, markedly enhances AC activity and cAMP signaling, may be an ideal model in which to elucidate the features of a wide-spread an adaptive paradigm that must become engaged in response to incessant chronic activation of cardiac AC/cAMP/PKA/Ca$^{2+}$ signaling. Specifically, concurrent with chronically increased AC activity within the young adult TG$^{AC8}$ sinoatrial node (SAN) and LV, (*Moen et al., 2019*; *Lipskaia et al., 2000*) heart rate (HR), measured via telemetry in the awake, unrestrained state, increased by approximately 30%, persisting in the presence of dual autonomic blockade; *Moen et al., 2019* and LV EF is also markedly increased in TG$^{AC8}$ (*Mougenot et al., 2019*). Thus, the cardiac phenotype of the adult TG$^{AC8}$ mimics cardiac responses to AC/cAMP/PKA/Ca$^{2+}$ sympathetic autonomic input during strenuous, acute exercise, but this stressful state persists incessantly (*Moen et al., 2019*). Although this continual high cardiac load imposes a severe chronic stress on the heart, which might be expected to lead to near term heart failure and demise (*Liu et al., 2020*; *Zhang et al., 2005*), adult TG$^{AC8}$ mice maintain this remarkable, hyperdynamic cardiac phenotype for up to about a one year of age, (*Lipskaia et al., 2000*; *Georget et al., 2002*; *Georget et al., 2003*; *Mougenot et al., 2019*) when signs of CHF begin to develop and cardiomyopathy ensues (*Mougenot et al., 2019*).

We hypothesized, (1) that a panoply of intrinsic adaptive mechanisms become *concurrently* engaged in order to protect the TG$^{AC8}$ heart during the incessant, high level of cardiac work in several months, and (2) that, some of these mechanisms are those that become activated in the endurance, trained heart, including shifts in mechanisms of energy generation, enhanced protein synthesis and quality control, and increased defenses against reactive O$_2$ species (ROS) and cell death (*Hanahan and Weinberg, 2000*; *Hanahan and Weinberg, 2011*). We reasoned that a discovery bioinformatics approach, in conjunction with deeper phenotypic characterization of the TG$^{AC8}$ LV, would generate a number of testable hypotheses about the characteristics of some of these mechanisms utilized to sustain this adaptive paradigm of heart performance and protection in response to severe, chronic, adenylyl cyclase-induced cardiac stress. To this end, we performed unbiased, RNASEQ and proteomic analyses of adult TG$^{AC8}$ and WT LVs, and selectively validated genotypic differences in numerous transcripts and proteins. Our results delineate a pattern of consilient adaptations that becomes activated within the TG$^{AC8}$ heart in response to chronically increased AC/cAMP/PKA/Ca$^{2+}$-signaling and offers

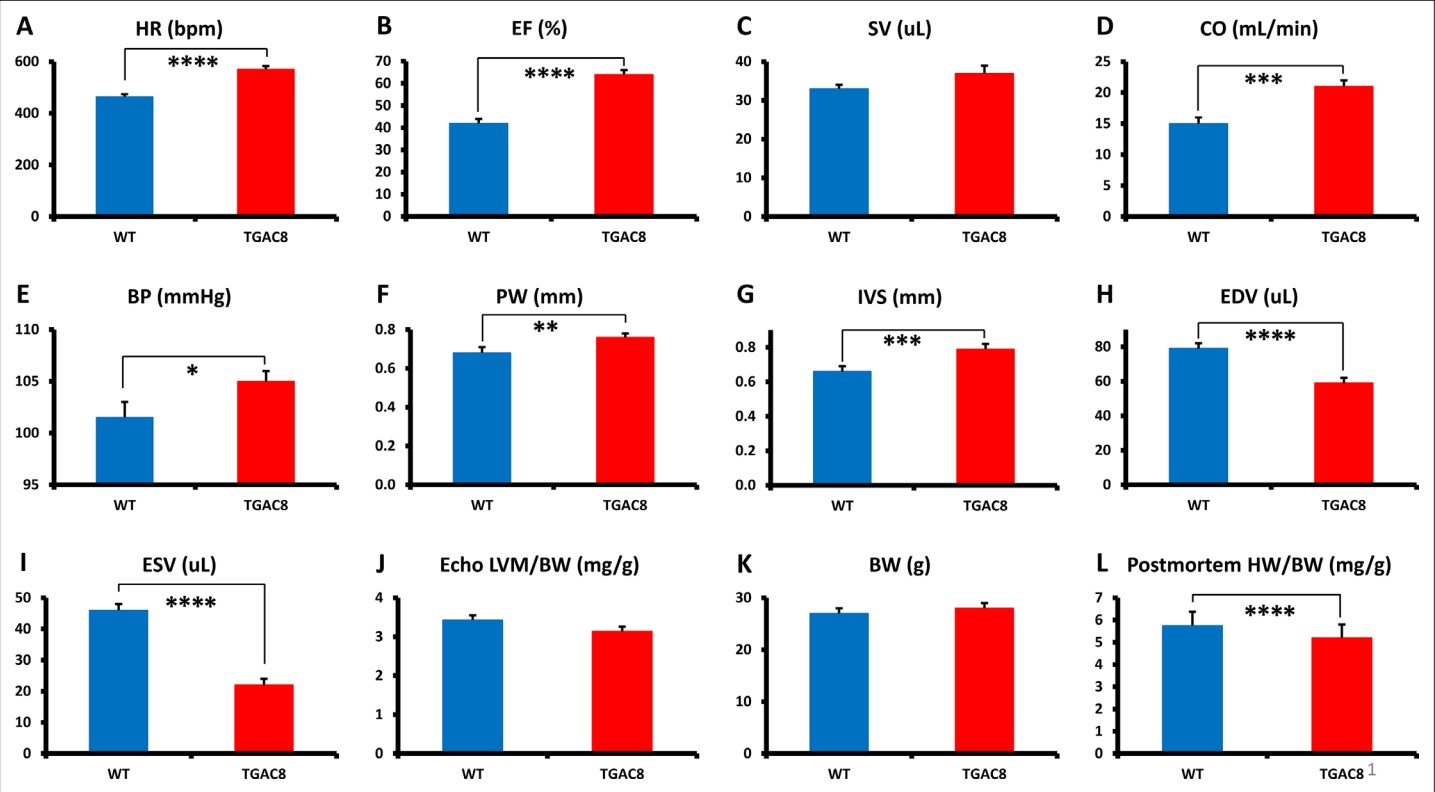

**Figure 1.** Echocardiographic parameters and body weight of TG[AC8] and WT mice. (**A–K**), Echocardiographic parameters (N=28 for TG[AC8]; WT = 21); (**L**) heart weight/body weight at sacrifice (N=75 for TG[AC8] and N=85 for WT). See ***Supplementary file 1a*** for additional Echo parameters. Data are presented as Mean± SEM. The statistical significance is indicated by * p<0.05, ** p<0.01, **** p<0.0001 (t test).

The online version of this article includes the following figure supplement(s) for figure 1:

**Figure supplement 1.** Representative images of echocardiograms.

numerous testable hypotheses to further define the details of mechanisms that underlie what we will show here to be a remarkable adaptive heart paradigm.

## Results

### Cardiac structure and performance

We performed echocardiography and cardiac histology for in-depth characterization of the TG[AC8] heart structure and function. Representative echocardiograms of TG[AC8] and WT are illustrated in *Figure 1—figure supplement 1*, selected Echo parameters are shown in *Figure 1*, (and a complete listing of parameters is provided in *Supplementary file 1a*). Both EF and HR were higher in TG[AC8] than in WT (*Figure 1A and B*), confirming prior reports (*Moen et al., 2019*; *Mougenot et al., 2019*). Because stroke volume did not differ by genotype (*Figure 1C*), cardiac output was elevated by 30% in TG[AC8] (*Figure 1D*) on the basis of its 30% increase in HR. Arterial blood pressure was only mildly increased in TG[AC8] averaging 3.5 mmHg higher than in WT (*Figure 1E*).

A sustained high cardiac workload is usually expected to result in an increase of a total LV mass, *Vega et al., 2017*; *Dorn, 2007*; *Maillet et al., 2013* that is, cardiac hypertrophy. Although both the LV posterior wall and inter-ventricular septum were thicker in TG[AC8] vs WT (indicative of an increased LV wall biomass; *Figure 1F and G*) the LV cavity volume was markedly reduced (*Figure 1H, I*), and the echo derived **total** LV mass did not differ by genotype (*Figure 1J*). Postmortem measurements indicated that although the body weight did not differ between TG[AC8] and WT (*Figure 1K*), the heart weight/body weight (HW/BW) was actually modestly reduced in TG[AC8] vs WT (*Figure 1L*).

Pathological hypertrophy markers, for example, β-myosin heavy chain (MYH7), ANP (NPPA) or BNP (NPPB) were not increased in TG[AC8] vs WT by Western Blot (WB) (*Figure 2A*); however, α skeletal

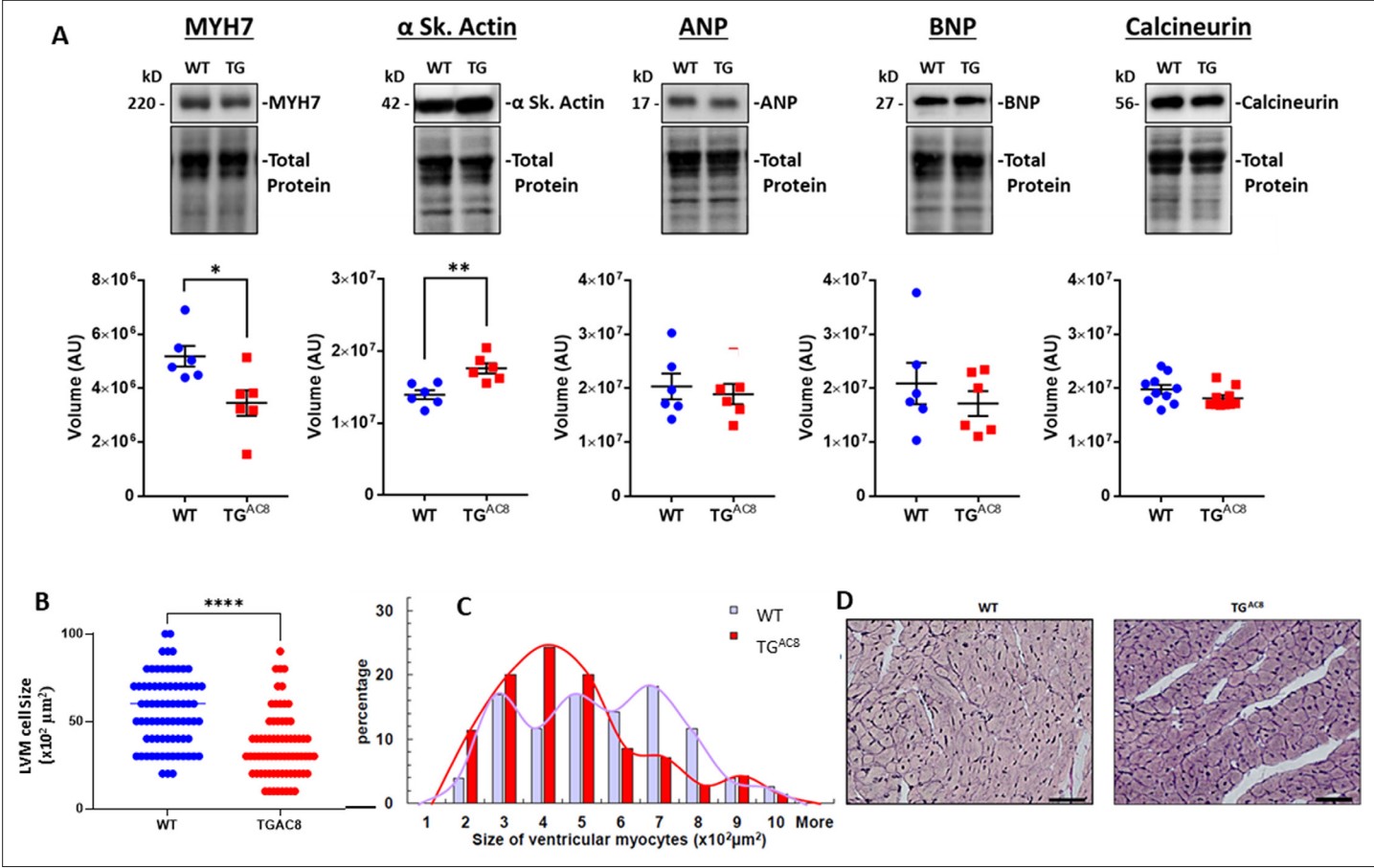

**Figure 2.** Expression of pathologic hypertrophy markers and cardiac myocyte sizes in LV of TG[AC8] and WT mice. (**A**) WB of Pathologic hypertrophy markers, (**B, C**) cardiac myocyte cross-sectional areas and distributions of cardiac myocyte sizes in TG[AC8] and WT hearts (cardiomyocytes were counted on left free wall at the middle level sections from four animals in each group; 37 high power fields from WT mice and 33 high power fields from TG[AC8] mice; 77 cells from WT mice and 70 cells from TG[AC8] mice were analyzed) Data are presented as Mean± SEM. The statistical significance is indicated by * p<0.05, ** p<0.01, (two-tailed t test)., (**D**) Representative LV sections depicting cardiac myocyte diameters (scale bar 100 μm).

The online version of this article includes the following source data and figure supplement(s) for figure 2:

**Source data 1.** Raw unedited blot images and uncropped blot images of MYH7, α Sk.

**Figure supplement 1.** Representative LV sections, labeled with picrosirius red (**A**) and (**B**) average collagen density (picrosirius red labeling) in TG[AC8] vs WT LV.

actin, commonly considered to be a pathologic hypertrophy marker, was increased in TG[AC8] vs WT (**Figure 2A**). Calcineurin (PP2B), which activates the hypertrophic response by dephosphorylating nuclear factor of activated T cells (NFAT), **Wilkins and Molkentin, 2004** did not differ in WB of TG[AC8] vs WT (**Figure 2A**).

## LV histologic analysis

The average LV cardiac myocyte size was smaller in TG[AC8] compared to WT (**Figure 2B and C**), and LV myocyte size distribution was different by genotype (**Figure 2D**). LV collagen content in TG[AC8] was not increased vs WT (**Figure 2—figure supplement 1**).

To address the issue of DNA synthesis within LV cardiac myocyte we loaded EdU for 28 days. EdU-labeled nuclei, detected in both TG[AC8] and WT whole mount ventricular preparations were randomly scattered throughout the LV from the mitral annulus to the apex. LV myocyte nuclei, however, were rarely EdU labeled. Rather nearly all EdU labeling was detected in small interstitial cells that expressed vimentin or in cells that enclosed the capillary lumina (**Figure 3A–G**), suggesting that EdU was also incorporated within the DNA of endothelial cells in both WT and TG[AC8]. However, total nuclear EdU labeling was 3-fold higher in TG[AC8] than in WT **Figure 3H**.

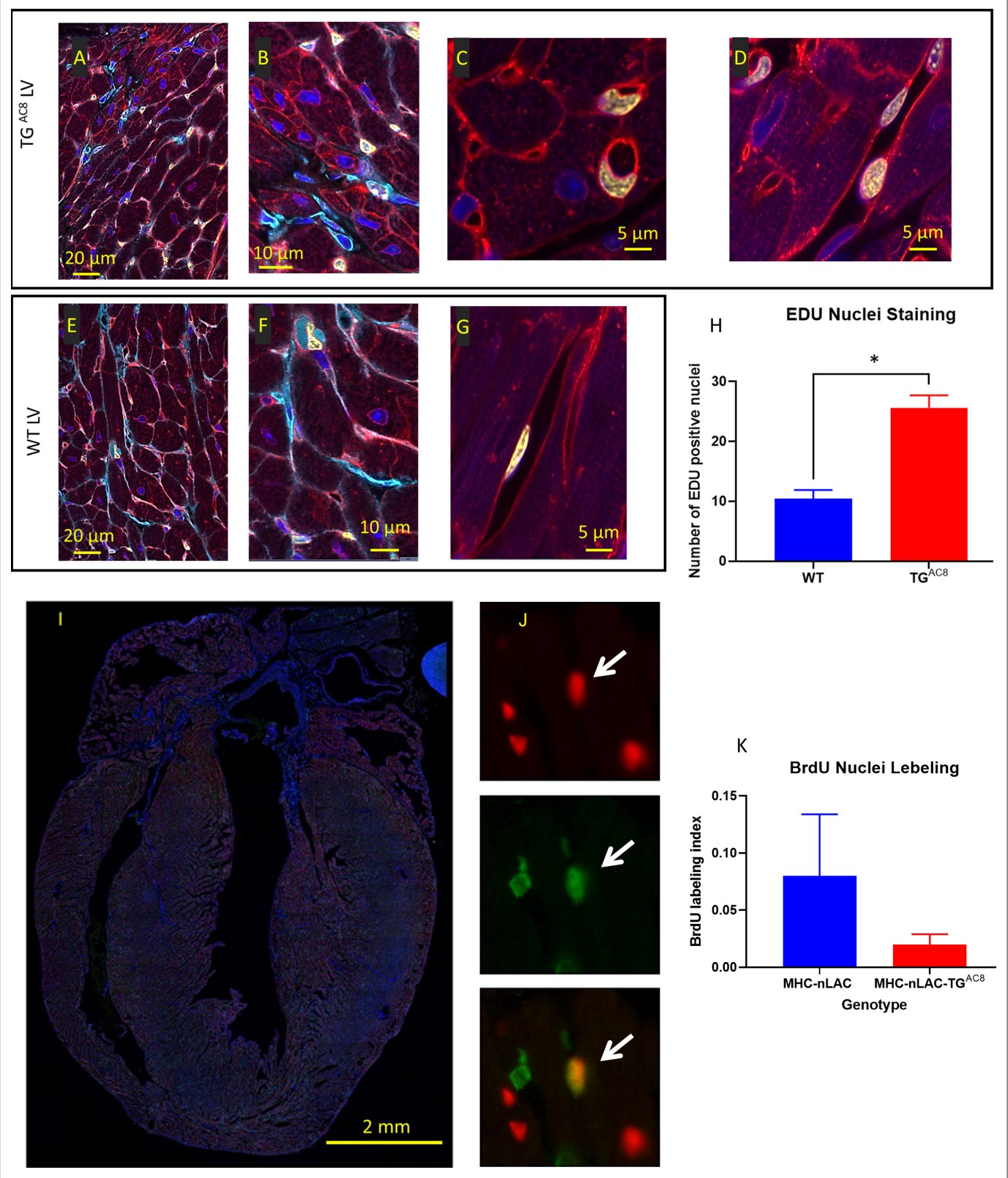

**Figure 3.** EdU labeling of cardiac tissue. (**A–G**) representative examples of confocal images (400 x) of LV WGA (red), vimentin (Cyan), EdU (yellow), DAPI (blue), (**H**) average number of EdU-labeled nuclei positive field counts in LV TG^AC8 vs WT (N=3 mice in each genotype). Data are presented as Mean± SEM. The statistical significance is indicated by * p<0.05, (t test). Detection of cardiomyocyte S-phase activity. (**I**) Section from the heart of a TG^AC8, MHC-nLAC double-transgenic mouse subjected to 12 days of BrdU treatment. The section was processed for β-galactosidase (to identify cardiomyocyte

*Figure 3 continued on next page*

*Figure 3 continued*

nuclei, red signal) and BrdU (to identify DNA synthesis, green signal) immune reactivity, and then counterstained with Hoechst (which stains all nuclei, blue signal). (**J**) Example of an S-phase cardiomyocyte nucleus detected with this assay. The upper panel shows β-galactosidase immune reactivity (red channel), the middle panel shows BrdU immune reactivity (green channel), and the lower panel shows a red and green color combined image of the same field. The arrow identifies an S-phase cardiomyocyte nucleus, as evidenced by the overlay of β-galactosidase and BrdU immune reactivity, which appears yellow in the color combined image. (**H**) Graph representing S-phase activity in the TG^AC8, MHC-nLAC double-transgenic vs. the MHC-nLAC single transgenic animals (mean +/- SEM, p=0.315; 5 mice per genotype and 3 sections per mouse were analyzed).

To monitor cardiomyocyte S-phase activity, TG$^{AC8}$ mice were crossed with MHC-nLAC mice (which express a nuclear-localized β-galactosidase reporter under the transcriptional regulation of the mouse α-cardiac MHC promoter; these mice are useful to identify cardiomyocyte nuclei in histologic sections) (*Soonpaa et al., 1994*). The resulting TG$^{AC8}$, MHC-nLAC double-transgenic mice and MHC-nLAC single-transgenic mice were identified and sequestered. At 28-to-30 days of age, the mice were administered BrdU via drinking water (0.5 mg/ml, changed every 2nd day) for a total of 12 days. There was no difference in the level of ventricular cardiomyocyte S-phase activity in the TG$^{AC8}$, MHC-nLAC double-transgenic vs. the MHC-nLAC single transgenic animals (*Figure 3H–K*).

Thus, the adult TG$^{AC8}$ heart has a hyper-dynamic LV with thicker walls, harboring not only an increased number of small **cardiac** myocytes, but also an increased number of small interstitial cells and endothelial cells with increased EdU labeling vs WT. The LV cavity volumes at both end-diastole and end-systole were markedly reduced, and LV EF was markedly increased in TG$^{AC8}$ vs WT. But, neither **total** LV mass, nor collagen content were increased in TG$^{AC8}$ vs WT and the profile of pathologic cardiac hypertrophy markers in TG$^{AC8}$ was absent.

## AC/cAMP/PKA/Ca$^{2+}$ signaling

Given that the transgene in our study was an AC, we next focused on expected differences in AC/cAMP/PKA/Ca$^{2+}$ signaling in TG$^{AC8}$ vs WT. Immunolabeling of single LV myocytes showed that AC8 expression was markedly increased (by 8–9-fold in TG$^{AC8}$ vs WT), and AC activity (measured in membranes isolated from LV tissue) in TG$^{AC8}$ was 50% higher than that in WT (*Figure 4A–C*). Expression of the PKA catalytic subunit was increased by 65.6%, and expression of the regulatory subunit was decreased by 26.7% in TG$^{AC8}$ by WB (*Figure 4D and E*). PKA activity was increased by 57.8% in TG$^{AC8}$ vs WT (*Figure 4F*).

To search for mechanisms that increase contractility within cardiac myocytes, we examined the expression of selected proteins downstream of PKA signaling that are involved in excitation/Ca$^{2+}$ release/contraction/relaxation. Western blot analysis revealed that a number of proteins that determine increase cardiac myocyte performance were upregulated in TG$^{AC8}$ vs WT, including αMHC (MYH6 by 18.7%), SERCA2 (ATP2A2 by 62.0%), L-type Ca Channel (Cav1.2, by 61.8%), and NCX1 (AKA SLC8A1) by 117.6%, and CaMKII by 35.0% (*Figure 4G–K*). Immunolabeling of total RyR2 was increased by 66% in TG$^{AC8}$ vs WT (*Figure 4L*). This pattern of increased protein expression is consistent with increased Ca$^{2+}$ flux into and out of cardiac myocytes, an increased Ca$^{2+}$ cycling between SR and cytosol, and increased myosin cross-bridge kinetics during heart contraction in TG$^{AC8}$ vs WT.

Thus, as would be expected in the context of markedly increased transcription of AC type VIII, AC and PKA protein levels and activities, and levels of proteins downstream of PKA signaling, were markedly increased and associated with a chronic, marked increase in LV performance.

## Protein synthesis, degradation, and quality control

Because PKA signaling-driven increased cardiac work is known to be associated with increased protein synthesis, *Pinson et al., 1993*; *Yamazaki et al., 1997* we next compared the rate of protein synthesis in TG$^{AC8}$ and WT LV lysates. Despite the absence of increase of total LV mass, the rate of protein synthesis was 40% higher in TG$^{AC8}$ than in WT (*Figure 5A*). WB analysis indicated that expression or activation of p21Ras, p-c-Raf, MEK1/2, molecules downstream of PKA signaling that are implicated in protein synthesis, were increased in TG$^{AC8}$ vs WT (*Figure 5B–C*). The transcription factor CREB1, involved in PKA signaling directed protein synthesis was increased by 58.1% in TG$^{AC8}$ vs WT in WB analysis (*Figure 5D*). Expression of CITED4 (family of transcriptional coactivators that bind to several proteins, including CREB-binding protein [CBP]) was increased by 51% in TG$^{AC8}$ vs WT in WB analysis (*Figure 5E*). Expression of protein kinases, that are required for stress-induced activation of CREB1,

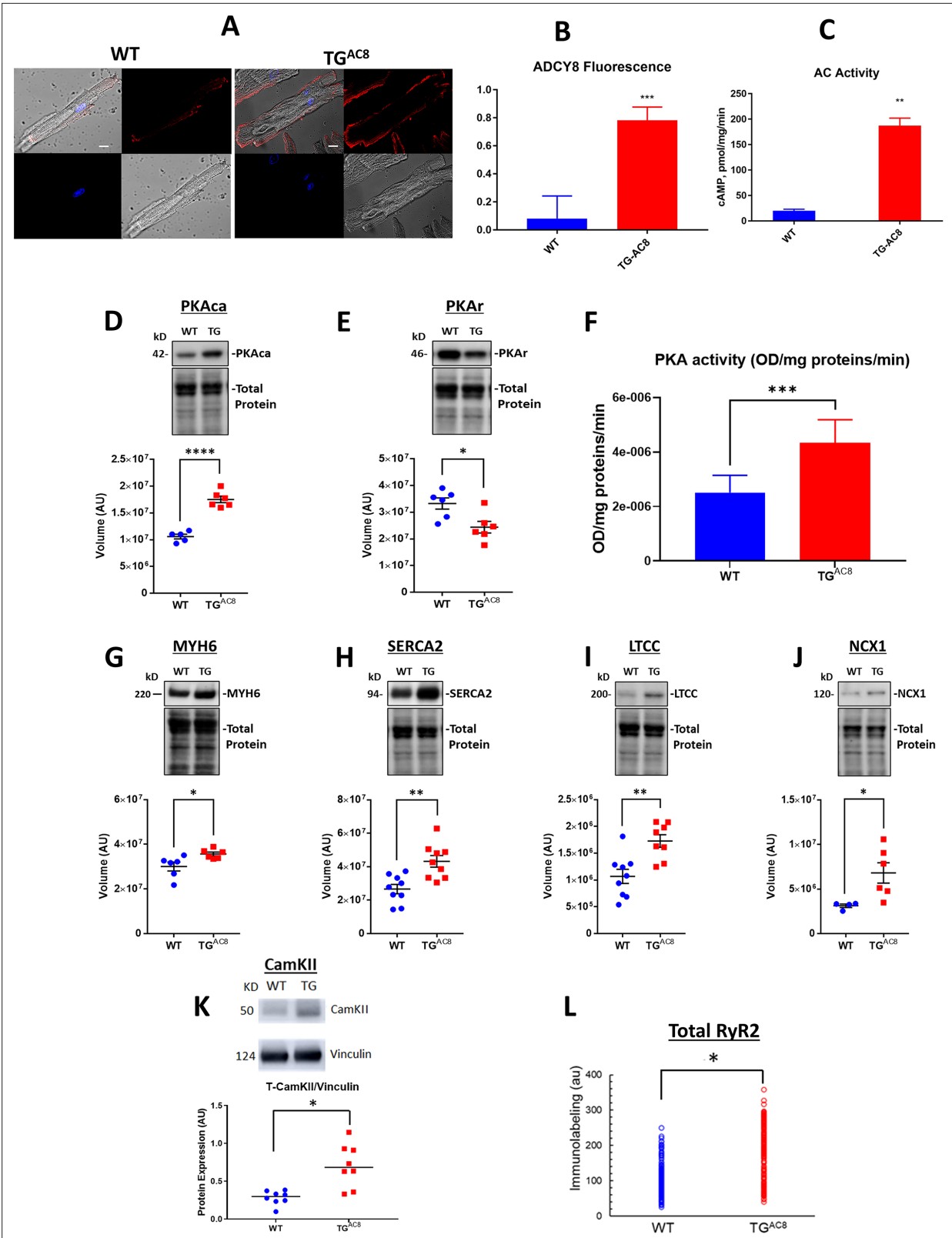

**Figure 4.** Expression of molecules involved in AC/cAMP/PKA/Ca²⁺ signaling in LV of TG^AC8 and WT mice. (**A**) Representative examples of ADCY8 immunolabeling in TG^AC8 and WT LV cardiomyocytes (scale bar 10 μm), (**B**) Average AC8 fluorescence in LV cardiomyocytes (N=3 animals per group; 60 cells for each genotype were analyzed. Data are presented as Mean± SEM. The statistical significance is indicated by **** p<0.0001 (two-tailed t test).) and (**C**) Average AC activity in TG^AC8 vs WT (N=3 per group), (**D, E**) Expression levels of PKA catalytic and regulatory subunits, and (**F**) PKA activity

*Figure 4 continued on next page*

*Figure 4 continued*

in TG^AC8 vs WT (N=8 per group), (**G–M**) Western Blot analysis of selected proteins involved in excitation - Ca release – contraction-relaxation coupling TG^AC8 vs WT LV. (**L**) RyR2 immunolabeling. Antibodies employed are listed in supplemental methods. (N=199 WT cells and 195 TG^AC8 cells each from 3 mice). Data are presented as Mean± SEM. The statistical significance is indicated by * p<0.05, ** p<0.01, *** p<0.001,**** p<0.0001 (two-tailed t test).

The online version of this article includes the following source data for figure 4:

**Source data 1.** Raw unedited blot images and uncropped blot images of PKAca, PKAr, MYH6, SERCA2, LTCC, NCX1, CamKII.

MSK1 and MNK1, direct substrates of MAPK, were increased by 40% and 48% in TG^AC8 vs WT, respectively (*Figure 5F and G*).

## Autophagy

Proteasome activity within LV lysates was increased in TG^AC8 vs WT (*Figure 6A*). However, although there was a significant increase in the amount of soluble misfolded proteins in TG^AC8 vs WT (*Figure 6B*), insoluble protein aggregates did not accumulate within the TG^AC8 LV (*Figure 6C*). This suggest that an increase in micro-autophagy of the TG^AC8 LV circumvents the potential proteotoxic stress of aggregated protein accumulation, which can negatively impact cardiac cell health and function. A 24% increase in the expression of HSP90α (*Figure 6D*) in TG^AC8 suggested that chaperone-mediated autophagy was also involved in preventing an accumulation of insoluble protein aggregates.

We reasoned that another type of protein quality control, macro-autophagy, might also be activated in TG^AC8 vs WT. Indeed, protein levels of several members of the autophagy machinery were increased in TG^AC8 vs WT: both ATG13, a factor required for autophagosome formation and mitophagy,

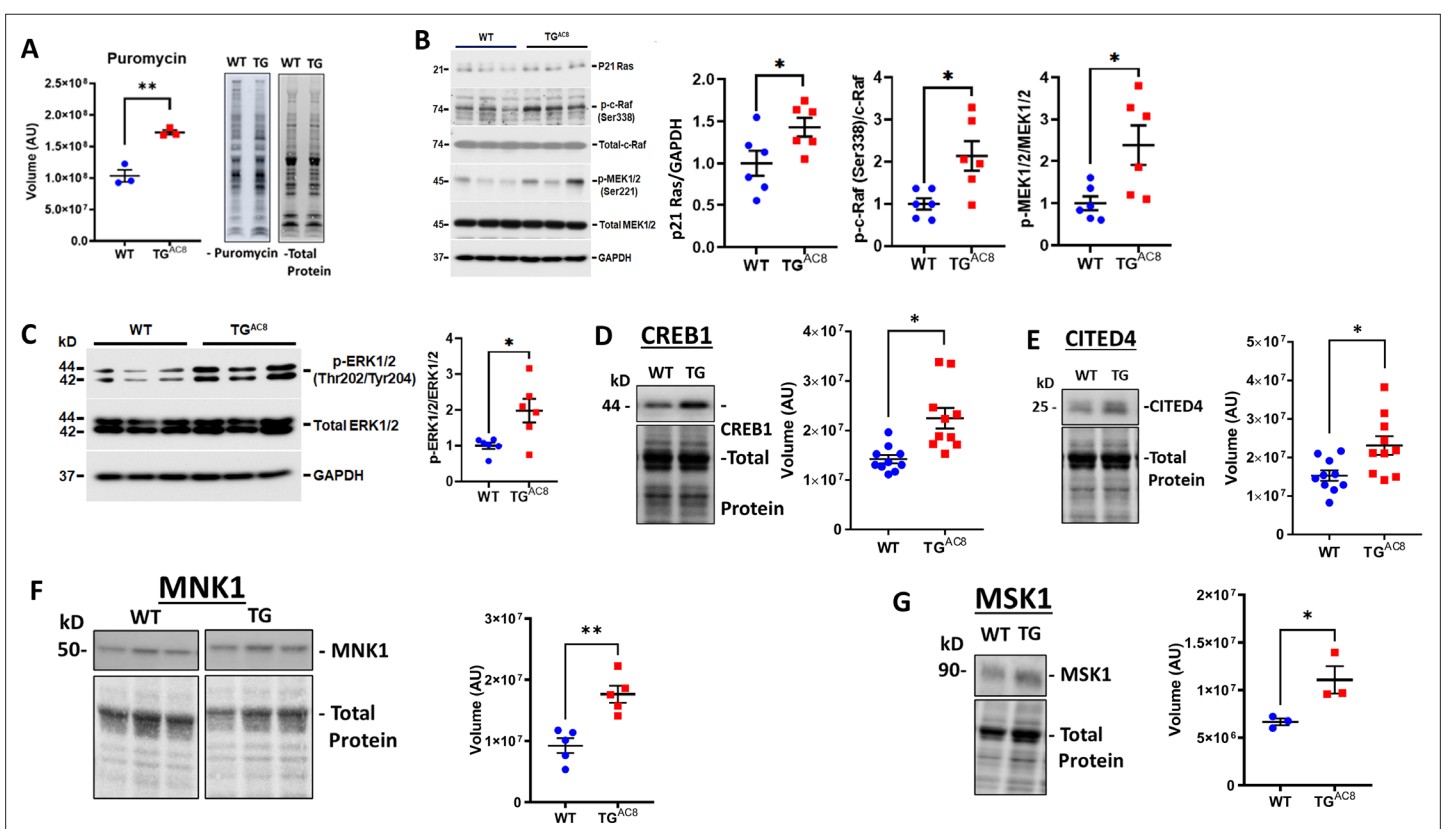

**Figure 5.** Expression of proteins,involved in protein synthesis, degradation, and quality control in LV of TG^AC8 and WT mice. (**A**) Rate of global protein synthesis and (**B–G**) mechanisms downstream of PKA signaling involved in protein synthesis in the TG^AC8 and WT. Data are presented as Mean± SEM. The statistical significance is indicated by * p<0.05, ** p<0.01,(two-tailed t test).

The online version of this article includes the following source data for figure 5:

**Source data 1.** Raw unedited blot images and uncropped blot images of Puromycin, CITED4, CREB1, MNK1, MSK1, GAPDH, pMEK1/2, p21 Ras, p-c-Raf, total Raf, total Mek 1/2, p-ERK 1/2 and total ERK 1/2.

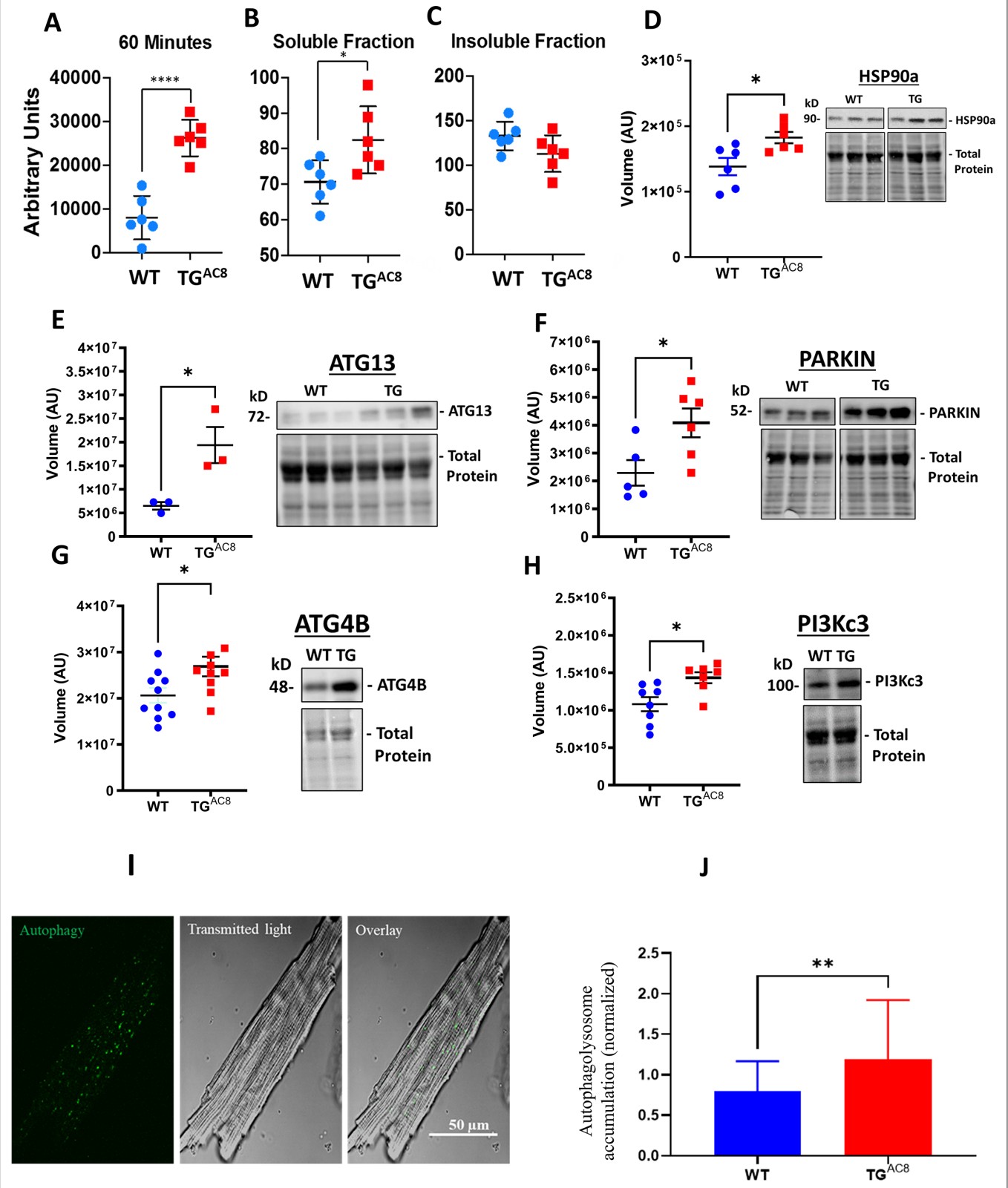

**Figure 6.** Proteosome activity assay, accumulated proteins in soluble and insoluble fractions and expression levels of selected proteins involved in the autophagy process in LV of TG^AC8 and WT mice. (**A**) Proteosome activity assay and (**B, C**) accumulated proteins in soluble and insoluble fractions of LV lysates in TG^AC8 vs WT. (**D**) WB of HSP90 in TG^AC8 and WT, (**E–H**) Expression levels of selected proteins involved in the autophagy process (Data are presented as Mean± SEM. The statistical significance is indicated by * p<0.05, **** p<0.0001 (two-tailed t test).). ;(**I, J**) Autophagolysosome

*Figure 6 continued on next page*

*Figure 6 continued*

accumulation is enhanced in AC8 mice (46 cells for each group; 3 aminals per genotype). Data expressed as Mean± SEM. The statistical significance is indicated by \*\*p<0.01, and t (p<0.01 in one-tailed t test).

The online version of this article includes the following source data for figure 6:

**Source data 1.** Raw unedited blot images and uncropped blot images of HSP90, ATG13, PARKIN, ATG4B, and PI3Kc3.

and the ubiquitin-protein ligase PARKIN, which protects against mitochondrial dysfunction during cellular stress, by coordinating mitochondrial quality control mechanisms to remove/replace dysfunctional mitochondrial components, were significantly increased in TG[AC8] vs WT (*Figure 6E and F*). Furthermore, protein levels of the cysteine protease ATG4B, involved in processing, lipidation/delipidation (conjugation/removal of phosphatidylethanolamine) and insertion of MAP1LC3, GABARAPL1, GABARAPL2 and GABARAP into membranes during autophagy and endocytosis, were also increased (*Figure 6G*). In addition, the catalytic subunit of the PI3Kc3, involved in initiation and maturation of autophagosomes and in endocytosis, was also significantly higher in the TG[AC8] vs WT (*Figure 6H*). Finally, direct measurements in single LV myocytes showed that autophagolysosomes were appreciably increased in TG[AC8] vs WT, indicating that autophagy (mitophagy) was activated to a greater extent in the TG[AC8] vs WT. (*Figure 6I and J*).

## Mitochondrial structure

We employed transmission electron microscopy (TEM) to directly visualize ultrastructural details of the mitochondria and cardiac myofibers within the LV of TG[AC8] and WT. Representative panoramic electron micrographs of LV cardiac muscle fibers and mitochondria in TG[AC8] and WT are illustrated in *Figure 7A and B*. Cardiac myocytes, presenting a very distinctive morphology with high content of myofibrils, a large number of high-electron dense mitochondria and several capillaries surrounding cardiac myocytes, are depicted (see white arrows). Cardiac myocyte ultrastructure is depicted in *Figure 7* panels **C** and **E**, for WT mice, and in panels **D** and **F**, for TG[AC8]. Asterisks show swollen, disrupted mitochondria with lighter cristae compared to the surrounding healthy mitochondria. *Figure 7* Panels **G-J** present quantitative stereological analyses of normal and damaged mitochondria. Although there was a mild increase in the number of damaged mitochondria (0.3 %), and in the percent of cell volume occupied by damaged mitochondria (0.4%) in TG[AC8], the numbers of healthy mitochondria and the percent of cell volume occupied by healthy mitochondria did not differ between TG[AC8] and WT. Nevertheless, the presence of mitochondrial deterioration at a young age is uncommon and may be further evidence for enhanced cleaning and recycling mechanisms such as autophagic signaling, (*Figures 6 and 7*).

## Mitochondrial fitness

The healthy functioning and survival of cardiac myocytes during severe, chronic myocardial stress requires close coordination of survival mechanisms and numerous mitochondrial functions that require a high level of mitochondrial fitness (*Zorov et al., 2000*; *Juhaszova et al., 2004*; *Zorov et al., 2014*; *Aon et al., 2021*). The mitochondrial permeability transition pore (mPTP) is a key regulator of mitochondrial functions, including energy metabolism (e.g. with the pore performing as a 'safety valve', opening transiently and reversibly, to prevent: (1) the excessive accumulation of certain regulatory species, such as $Ca^{2+}$; and (2) bioenergetic byproducts/damaging reactive species, such as free radicals, from achieving toxic levels). The mPTP also regulates cell fate: enduring and irreversible pore opening, plays decisive mechanistic roles in mitochondrial and cell life vs death decisions during normal development or pathological stress (e.g. involving excess and damaging free radical exposure). Measurement of the pore susceptibility or resistance to being induced/opened can serve as a biomarker of mitochondrial fitness. *Figure 7* panels **M** and **N** shows that the mPTP ROS threshold did not differ in TG[AC8] vs WT, suggesting a comparable degree of mitochondrial fitness in both genotypes.

## ROS levels and NRF signaling

Given the incessantly elevated myocardial contractility and heart rate, and increased protein synthesis and quality control mechanisms in TG[AC8] vs WT, it might be expected that ROS levels are increased in TG[AC8]. To this end, we measured superoxide radical accumulation in isolated, perfused, isometrically

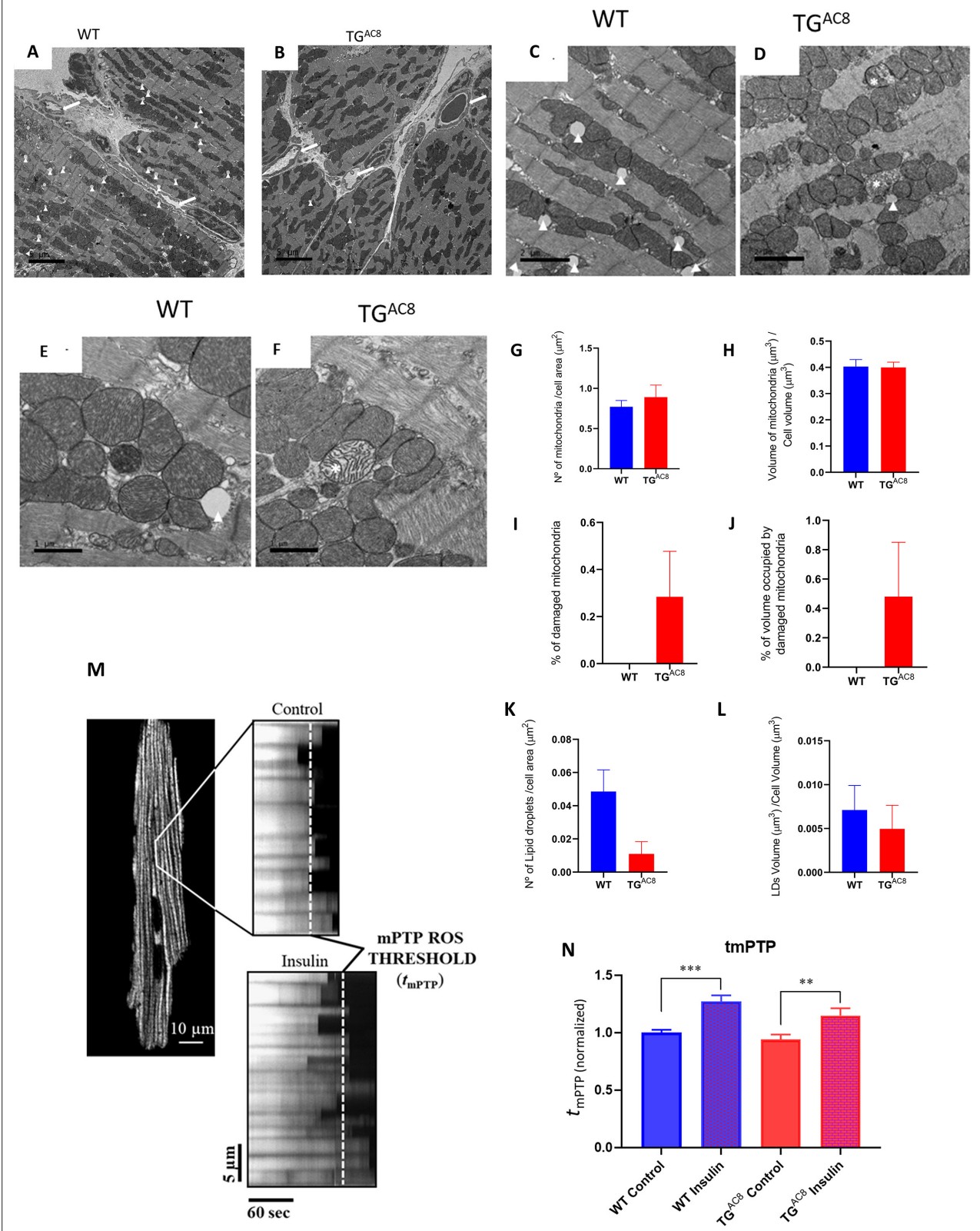

**Figure 7.** Mitochondrial structure and function. (**A, B**) Representative panoramic electron micrographs (white arrows depict capillaries surrounding cardiac myocytes) and (**C–F**) higher resolution images of LV cardiac muscle fibers and mitochondria in TG$^{AC8}$ and WT. White arrows depict lipid droplets; asterisks show swollen, disrupted mitochondria with lighter cristae compared to the surrounding, healthy mitochondria. (**G, H**) Average mitochondrial number of quantitative stereological analyses of normal mitochondrial number and volume, (**I, J**) damaged mitochondria, (**K, L**) number of lipid droplets

*Figure 7 continued on next page*

*Figure 7 continued*

per cell area and volume of lipid droplets. (**M, N**) mPTP-ROS threshold, measured in a single LV cardiac myocyte, did not differ between TG^(AC8) and WT mice. Insulin was employed as a positive control. (3 animals in each genotype; 29 cells - WT Control; 26 cells - WT insuline; 30 cells - TG^(AC8) control; 25 cells - TG^(AC8) insulin) (** p<0.01, *** p<0.001, one-way anova; Mean± SEM).

contracting TG^(AC8) hearts that maintained markedly enhanced cardiac workload observed in vivo (*Figure 8A*). A marked increased heart rate pressure product in TG^(AC8), suggests that myocardial oxygen consumption was increased vs WT, possibly due to increased coronary flow or increased oxygen extraction from the perfusate. Superoxide radical accumulation did not differ by genotype, suggesting that mechanisms to scavenge ROS are increased in TG^(AC8). Indeed, the level of NRF2 protein, a key regulator of ROS defense signaling was increased by 24% in TG^(AC8) vs WT (*Figure 8B*), suggesting that increased NRF signaling in the TG^(AC8) LV may be a factor that prevents the accumulation of superoxide ROS. Interestingly, NRF2 signaling, was one of the top enriched and activated signaling pathway in transcriptome and proteome bioinformatic analysis (see below and *Supplementary file 1i*).

## High-energy phosphates

Given the fact that increased protein synthesis and quality control mechanisms (*Figure 6*), maintenance of normal ROS levels (*Figure 8*) and the incessant high cardiac performance of the TG^(AC8) (*Figure 1*) require increased energy production, it is reasonable to assume that the total energy requirements of the TG^(AC8) LV are probably considerably higher than those in WT. It was important, therefore, to assess high-energy phosphate levels in TG^(AC8) and WT. A schematic of ATP-creatine energy system is depicted in *Figure 9A*. Steady state levels of ATP, phosphocreatine and the ATP: phosphocreatine assessed in vivo were maintained at the same level in TG^(AC8) as in WT (*Figure 9B–E*), suggesting that

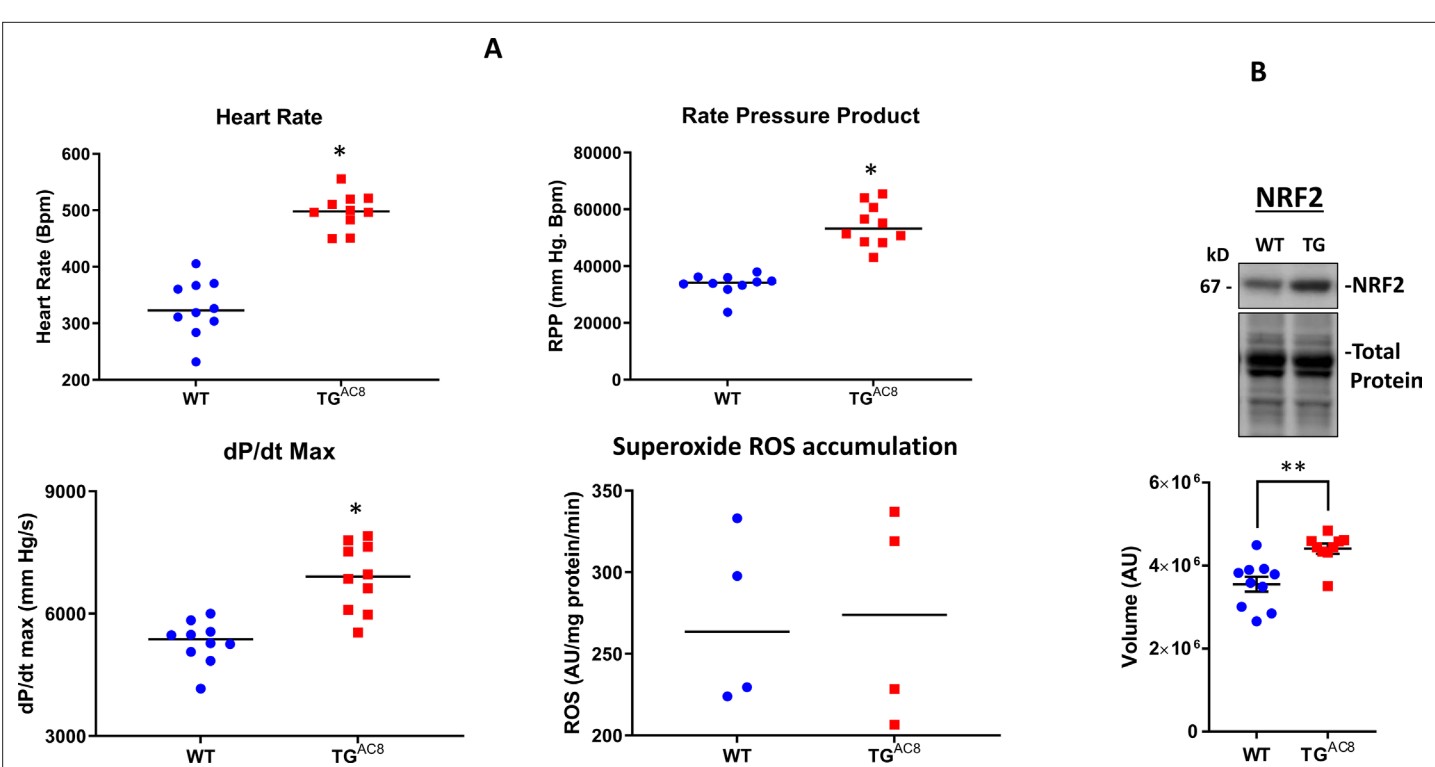

**Figure 8.** Detection of ROS levels and NRF signaling in LV of TG^(AC8) and WT mice. (**A**) LV performance and the rate of superoxide (ROS) generation in isolated working TG^(AC8) and WT hearts. (**B**) WB analysis of Nrf2 expression in LV TGA^(C8) vs WT. Differences between two groups were assessed by a t-test, and reported as Mean± SEM; * p<0.05; ** p<0.01.

The online version of this article includes the following source data for figure 8:

**Source data 1.** Raw unedited blot images and uncropped blot images of NRF2.

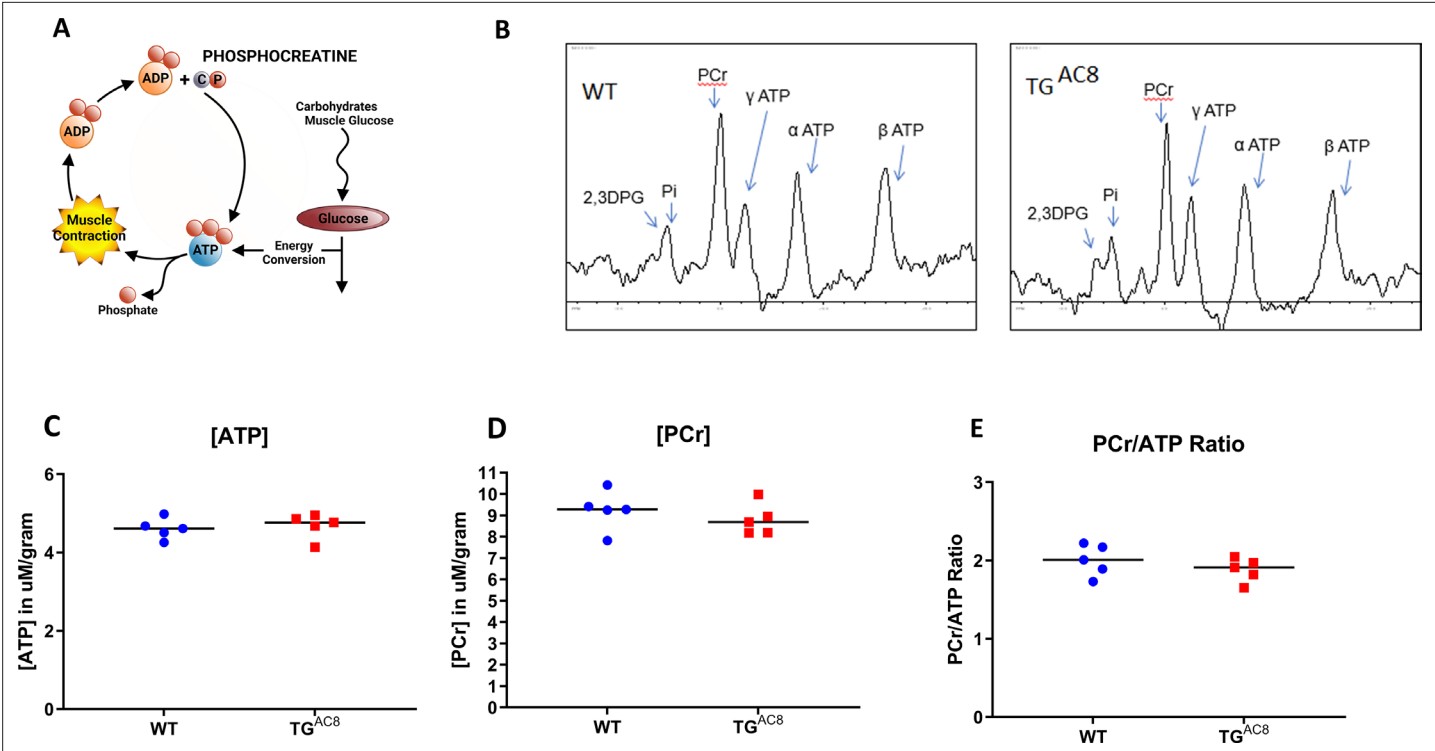

**Figure 9.** High-energy phosphates in TG^AC8 and WT hearts. (**A**) A schematic of ATP creatine energy system, (**B**) Representative P31 NMR spectra of TG^AC8 and WT hearts. (**C–E**) Average levels of ATP, PCr, and ATP/PCr in TG^AC8 and WT hearts derived from NMR spectra. Data are presented as Mean± SEM. The statistical significance is indicated by * $p < 0.05$ (two-tailed t test).

the rate of ATP production in the TG^AC8 LV is adequate to meet its increased energy demands, at least when animals rest.

## Unbiased RNASEQ and proteome analyses

We performed unbiased, RNASEQ and proteome analyses of adult TG^AC8 and WT left ventricles (LV) in order to realize facets of stress circuitry that are known to become activated in response to environmental stress, *Hanahan and Weinberg, 2000*; *Hanahan and Weinberg, 2011* beyond those identified in our experiments illustrated in *Figures 1–9*. We reasoned that taking advantage of the knowledge base within bioinformatic tools would also generate a number of testable hypotheses regarding the components of adaptive paradigm of heart performance and protection in response to the constitutive challenge of marked cardiac-specific overexpression of adenylyl cyclase type 8.

## LV transcriptome (RNASEQ)

RNA sequencing identified 11,810 transcripts (raw data submitted to GEO GSE205234) in LV lysates (*Figure 10A*, *Supplementary file 1b*); of these, 2323 were differentially expressed in TG^AC8 vs WT (*Figure 10A* and *Supplementary file 1c*): 1201 were significantly upregulated in TG^AC8 vs WT and 1117 were significantly downregulated. A volcano plot and heatmap of these transcripts are shown in *Figure 10—figure supplement 1A and C*. The transcript abundance of human *ADCY8* in TG^AC8 (LV) myocardium was more than 100-fold higher than the endogenous mouse isoform (*Figure 10—figure supplement 1E*).

## LV proteome

A total of 6834 proteins were identified (raw data submitted to MassIVE MSV000089554) in the LV proteome (*Figure 10A*, *Supplementary file 1d*); of these, 2184 were differentially expressed in TG^AC8 vs WT: 2026 were upregulated and 158 were downregulated in TG^AC8 (*Supplementary file 1e*). A Volcano plot and heatmap of the proteome are shown in the *Figure 10—figure supplement 1B and D*. The Small Proline Rich Protein 1 **Sprr1a,** which is linked to induction of protein synthesis (Figure

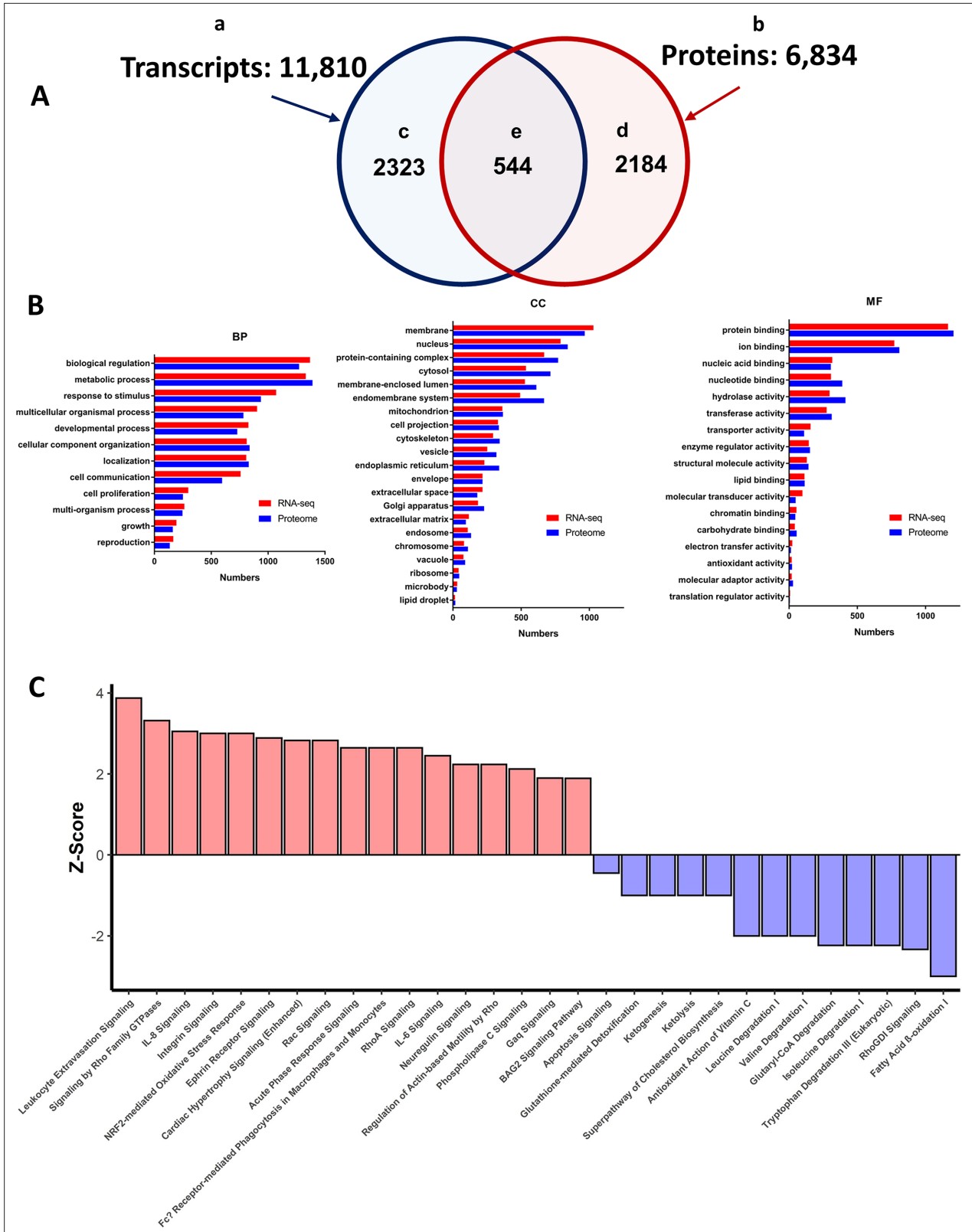

**Figure 10.** Summary of LV transcriptome and proteome analysis of TG[AC8] and WT. (**A**) Schematic of the total number of transcripts (subset 'a' – 11,810), and proteins (subset 'b' 6834), identified in LV lysates, the number of transcripts (subset 'c' 2323), and proteins (subset 'd' 2184), that differed by genotype, and number of identified transcripts and proteins that both differed by genotype (subset 'e' - 544). (**B**) WEBGESTALT analysis of the 2323 transcripts (Panel A subset 'c') and 2184 proteins (Panel A subset 'd') that significantly differed by genotype. Biological Processes (BP), Cell

*Figure 10 continued on next page*

*Figure 10 continued*

Compartment (CC), Molecular Functions (MF). (**C**) Top canonical signaling pathways differing in enrichment (-log10(pvalue) >1.3) and activation status by genotype in IPA analysis of transcripts and proteins.

The online version of this article includes the following figure supplement(s) for figure 10:

**Figure supplement 1.** Volcano plots and heat maps of transcripts and proteins.

**Figure supplement 2.** Spearman's correlations of Z-scores of IPA canonical pathways that significantly differed by genotype in transcriptome and proteome.

**Figure supplement 3.** IPA representation of top disease-related functions within the LV transcriptome and proteome of TG^AC8 and WT.

**Figure supplement 4.** Correlation plot of 544 identified molecules of which both transcripts and proteins differed by genotype.

**Figure supplement 5.** PROTEOMAP.

---

13, bottom panel), was the most abundantly overexpressed transcript **and** protein in the TG^AC8 **LV** (*Supplementary file 1i*). Immunolabeling of Sprr1a in single LV myocytes isolated from TG^AC8 is illustrated in *Figure 13—figure supplement 5*.

## Bioinformatic analyses of the *Total* transcriptome and proteome datasets

We employed online platforms WEB-based GEne SeT AnaLysis Toolkit (WebGestalt) and Ingenuity Pathway Analysis (IPA), for in depth analyses of the total 2323 expressed transcripts (*Figure 10A–a*) and the total 2184 (*Figure 10A–b*) proteins that differed by genotype.

In Gene Ontology (GO) analysis, WebGestalt classifies submitted molecules into three main categories, that is, Biological Process (BP), Cellular Component (CC) and Molecular Function (MF) (*Figure 10B*). Remarkably, transcriptome and proteome gene ontology (GO) terms that differed by genotype in response to overexpression of a single gene, AC Type 8, covered nearly all the biological processes and molecular functions within nearly all compartments of the TG^AC8 LV myocardium.

In order to discover genotypic differences in canonical signaling pathway enrichment, we performed QIAGEN Ingenuity Pathway Analysis (IPA) of the total number of transcripts (2,323) that were differentially expressed by genotype (*Figure 10A–c*) and the total number of proteins (2,184 *Figure 10A–d*). In IPA analyses 248 canonical pathways in the transcriptome analyses and 308 canonical pathways in the proteome differed by genotype p<0.05 in *enrichment* (listed in *Supplementary file 1f and g*). 154 pathways in transcriptome or proteome were *activated* by genotype (Z-score calculated) and Spearman's correlation coefficient $r_s$ was 0.57, p<0.0001 *Figure 10—figure supplement 2*, and showed moderate correlation between enriched and activated pathways in transcriptome and proteome. We performed IPA for identification of top Disease-related functions within the LV transcriptome and proteome of TG^AC8 and WT *Figure 10—figure supplement 3*.

Although 60 molecules were represented in only a single enriched pathway, 22 molecules were represented in 10 or more pathways, the highest among these being: MAP3K – (68 pathways); MAP2K2 – (51 pathways); and PLCG2 – (33 pathways) (*Supplementary file 1h*). Representation of several differentially expressed molecules in multiple canonical signaling pathways that differed by genotype is a plausible explanation accounting for the pattern within a TG^AC8 heart *of concentric*, concurrent pathway enrichment and activation (*Hanahan and Weinberg, 2000*; *Hanahan and Weinberg, 2011*).

The top enriched and activated or inactivated pathways are illustrated in *Figure 10C*. Of note, several pathways, for example, Nrf2 signaling, Integrin Signaling, and Cardiac Hypertrophy Signaling were among the top activated pathways that were highly enriched in TG^AC8 vs WT, whereas fatty acid β-oxidation, tryptophan degradation III, isoleucine degradation and glutaryl Co-A degradation were markedly suppressed in TG^AC8 vs WT (*Figure 10C*). Surprisingly, the top differentially regulated pathways between TG^AC8 and WT was 'leukocyte extravasation signaling' (*Figure 10C*).

## Bioinformatic analyses of transcripts and proteins that *both* significantly differed by genotype

544 transcripts and proteins **both** differed by genotype (*Figure 10A–e*, *Supplementary file 1i*). Of these 544, 339 (62.32%) were significantly **upregulated** in TG^AC8; and 99 (18.2%) were significantly

downregulated in TG^AC8. Thus, of the 544 transcripts and proteins that **both** significantly differed by genotype, 80.5% differed **concordantly** (in the same direction, *Figure 10—figure supplement 4*).

We next subjected these 544 molecules to IPA analysis. A total of 170 of the **same** enriched canonical pathways in the transcriptome and proteome differed by genotype. Of these, 118 pathways also differed by genotype and activation status (*Supplementary file 1j*).

In addition to IPA analyses we also used PROTEOMAP platform (*Liebermeister et al., 2014*) to visualize functional categories of the 544 proteins that differed by genotype (*Figure 10A–e*, *Figure 10—figure supplement 4*). PROTEOMAP displays a protein data set as functional trees (*Figure 10—figure supplement 5*), each consisting of a number of polygons, the areas of which reflect genotypic differences in protein abundances. Genotypic differences were most marked by increases in TG^AC8 of: ENVIRONMENTAL INFORMATION PROCESSING (informing on marked differences by **Rap1** and **PI3K-AKT** signaling pathways, harboring the marked increases in expression of ADCY8 and TNC (tenascin protein)); GENETIC INFORMATION PROCESSING (informing on PROTEIN **translation, folding, sorting and degradation** signaling pathways, harboring marked increases in RNF128, MMP2, and WFS1); and CELLULAR PROCESSES (informing on **vesicular transport** and **exosome**, harboring increased NCF1, LCP1, and CTSZ proteins).

PROTEOMAP also identified major genotypic changes in BIOSYNTHESIS AND CENTRAL CARBON METABOLISM (*Figure 10—figure supplement 5*), informing on genotypic differences in: **membrane transport,** harboring increased SLC4A2 (regulates intracellular pH, biliary bicarbonate secretion, and chloride uptake); **other metabolic enzyme proteins** (PDK3, SULF2, and ALPK2), and carbohydrate metabolism; and **glycolysis**, harboring reduced ALDH1B1.

## Regulatory networks centered on cAMP and protein kinase A signaling

Having confirmed that both AC8 protein and AC activity were markedly increased in TG^AC8 vs WT (*Figure 4B and C*) it was reasonable to infer that the multitude of genotypic differences that were identified in WebGestalt, IPA and PROTEOMAP (*Figure 10B*, *Figure 10—figure supplement 5*; *Supplementary file 1fg and h*) might be ultimately (directly or indirectly) linked to increased signaling driven by the high levels of AC activity. Three main downstream targets of cAMP generated by AC activity are: protein kinase A (**PKA**), guanidine-nucleotide exchange proteins activated by cAMP (**EPACs**), and cyclic nucleotide-gated ion channels (**CNGCs**). We found no evidence of activation for 2 of these 3 targets. In fact: (1) Neither CNGC subunits α 1–4, nor β 1–2 transcripts or proteins were identified in our omics analysis; (2) omics analysis (*Supplementary file 1b and d*) provided no evidence to suggest that cAMP directed **Epac** signaling was upregulated in the TG^AC8 LV. More specifically, transcripts of *Rapgef* 1 thru 6 were significantly **downregulated** in TG^AC8 vs WT (*Supplementary file 1c*) and RAPGEF 2, 3, and 5 proteins identified in proteomic analyses did not differ by genotype (*Supplementary file 1d*). On the contrary, we found clear evidence of activation of PKA. In fact, we had shown that PKA catalytic subunit and PKA activity are substantially higher in TG^AC8 vs WT (*Figure 4D and F*), and PKA signaling was among the top pathways increased and activated in IPA analysis. We therefore next focused the IPA knowledge base on the PKA complex as the center of a number of **interacting** pathways. We noticed that in addition to cAMP, another upstream regulator of PKA signaling, ITGA5, was increased in TG^AC8 vs WT in both the transcriptome and proteome (*Figure 11*), suggesting increased crosstalk between intra and extra cellular signaling.

It is well known, that PKA signaling activates a cascade of enzymes beginning with tyrosine hydroxylase (TH), that results in a production of catecholamines (*Figure 10—figure supplement 1A*). Among these enzymes, dopamine decarboxylase, which converts DOPA to dopamine, was increased in TG^AC8 vs WT (*Figure 11—figure supplement 1B*), dopamine b-hydroxylase, which converts dopamine to norepinephrine did not differ by genotype, and phenylethanolamine N-methyltransferase (PNMT), which converts norepinephrine to epinephrine was reduced in TGAC8 vs WT. In the context of these genotypic differences in enzyme levels LV tissue dopamine was increased and DHPG was reduced and NE was borderline reduced (p=0.08) (*Figure 11—figure supplement 1B*). Interestingly, as noted previously *Moen et al., 2019*, plasma levels of dopa and dopamine were also incresed in TG^AC8 vs WT, whereas norepinephrine, epinephrine, DOPAC, and DHPG were reduced.

To further investigate the role of PKA-signaling in the extreme TG^AC8 phenotype, we next established the protein interaction network centered on PKA (*Figure 11*). Gene families of these proteins ranged from transcription regulators, kinases, peptidases and other enzymes to transmembrane

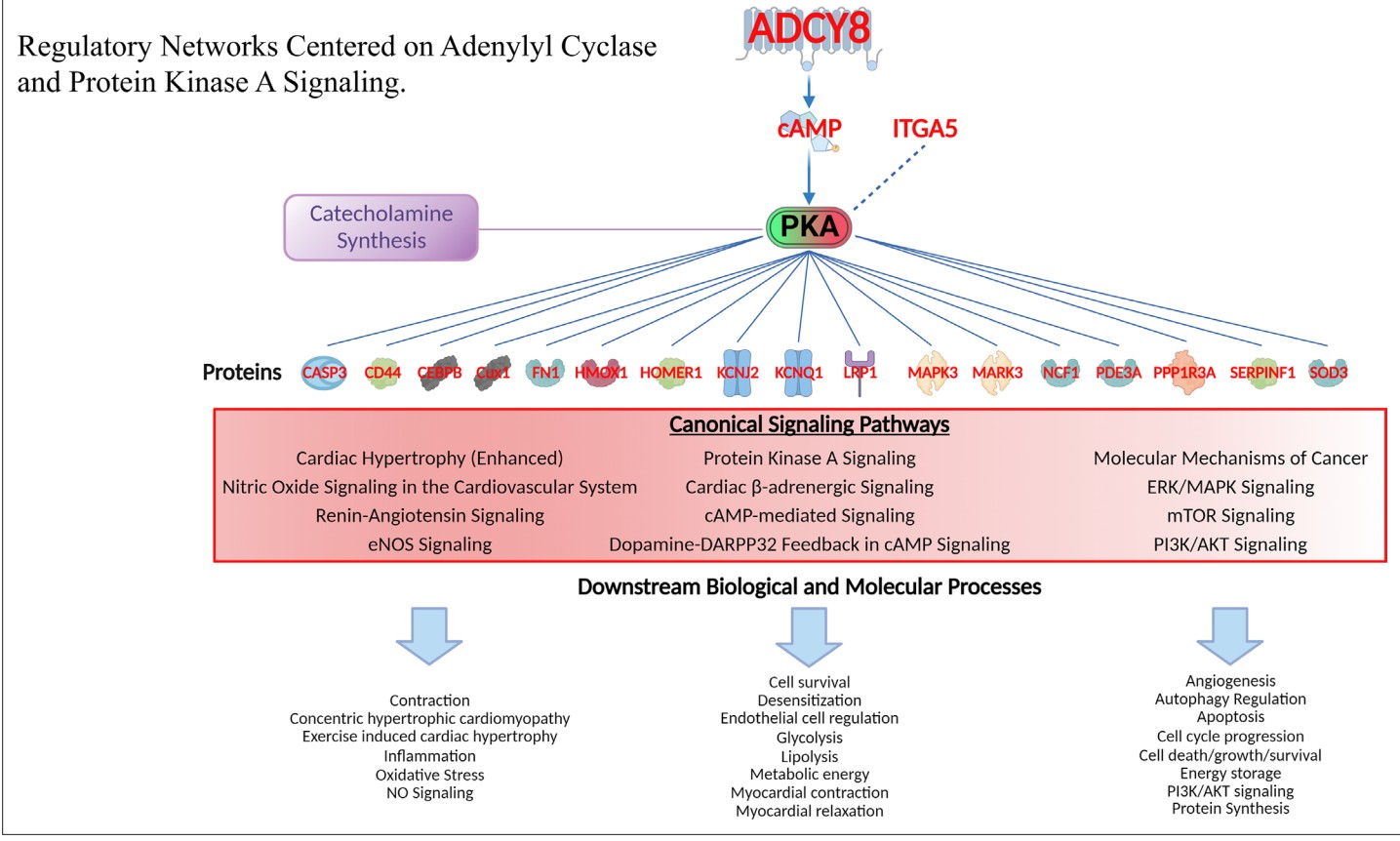

**Figure 11.** Regulatory networks centered on adenylyl cyclase and protein kinase A signaling. Gene families of proteins regulated by PKA ranged from transcription regulators, kinases, peptidases and other enzymes, transmembrane receptors, ion channels and other gene families. Canonical signaling pathways in which these proteins operate and downstream biological and molecular processes of proteins in these pathways are also displayed (lower part). See **Supplementary file 1k** for the full list of these downstream effects of PKA signaling.

The online version of this article includes the following figure supplement(s) for figure 11:

**Figure supplement 1.** Myocardial catecholamine levels in LV of TG[AC8] and WT.

receptors, ion channels and other protein families (CFH, CD44, HOMER1, and SERPINF1). Although CREB1 protein was not identified in proteome analyses, as noted, this canonical transcription factor and its phosphorylated form activated by PKA were markedly increased in WB analyses (**Figure 5D**, **Figure 13—figure supplement 1B**). Canonical signaling pathways in which these proteins operate and downstream biological and molecular processes of proteins in these pathways are listed in **Figure 11** (lower part). See **Supplementary file 1k** for the full list of these downstream effects.

## Shifts in metabolism

The enrichment or activation of the large number of signaling pathways in TG[AC8] vs WT (**Figure 11C** and **Figure 12**; **Supplementary file 1**) suggested that some aspects of metabolism differed by genotype. A detailed schematic of the genotypic differences in transcripts and proteins that related to metabolism circuits is illustrated in **Figure 12**, and WB validations of selected proteins are presented in **Figure 12—figure supplement 1**. Bioinformatic analyses suggested that nutrient-sensing pathways, including AMPK, insulin and IGF signaling, that induce shifts in aerobic energy metabolism were activated in TG[AC8] vs WT LV. Interestingly, LV glycogen staining was modestly increased in the TG[AC8] vs WT (**Figure 13—figure supplement 7**).

A close inspection of the bioinformatic analyses of central carbon metabolic processes (**Figure 12**) suggested that catabolism of glucose is markedly increased in TG[AC8] vs WT, while the fatty acid β oxidation pathway is concurrently reduced. An increase in myocardial glucose metabolism in TG[AC8] vs WT was suggested by increases in GLUT1, HK1, GSK3Β, GYS1, PFK, PGK1, PGAM1, and PKM. Of

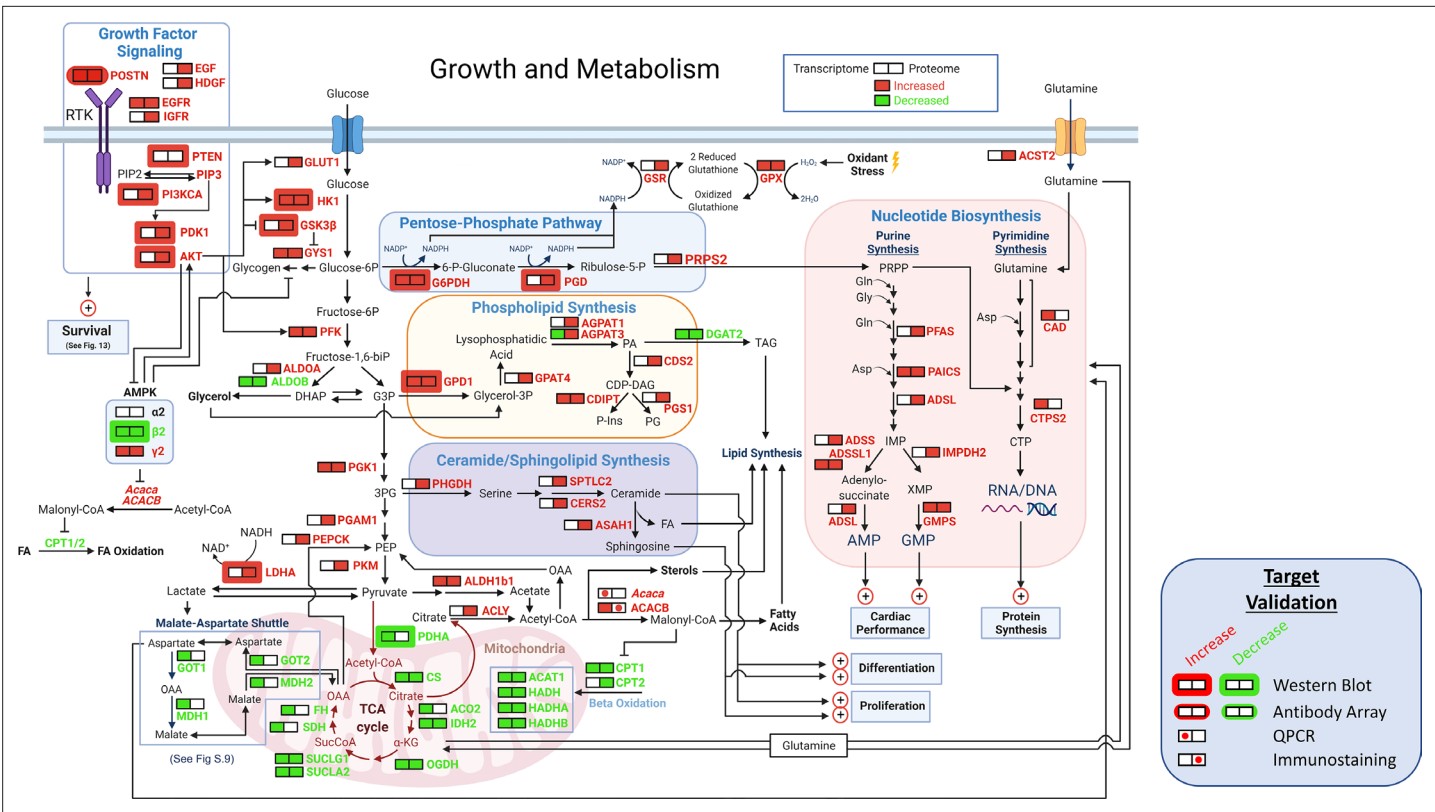

**Figure 12.** Growth and metabolism. A schematic of growth and metabolism circuitry based on signals derived from bioinformatic analyses of the transcriptome and proteome and on selected WBs. Catabolism of glucose is markedly increased in TG[AC8] vs WT as reflected in increased expression of Glut1, HK1, and PFK and other glycolytic enzymes, whilst fatty acid β oxidation pathway is concurrently reduced, as reflected in reduced expression of Cpt1, Cpt2, Acat1, Hadh, Hadha, and Hadhb. in TG[AC8] vs WT LV. Shifts in the transcription of genes and translation of proteins that are operative within PPS pathway, that is G6PDH, PGD and PRPS2 suggests that PPS is more highly activated in the TG[AC8] vs WT. The combination of increased expression of the glucose transporter, lactic acid dehydrogenase type A and the glutamine transporter in TG[AC8], suggests that, relative to WT, the TG[AC8] heart utilizes aerobic glycolysis to fulfill part of its energy needs. Enhanced growth factor and other PI3/AKT driven signaling processes increased in TG[AC8] vs WT are also known to be linked to aerobic glycolysis.

The online version of this article includes the following source data and figure supplement(s) for figure 12:

**Figure supplement 1.** WB analyses of selected proteins that mediate PIP3 kinase signaling and metabolism.

**Figure supplement 1—source data 1.** Raw unedited blot images and uncropped blot images of AKT2, GAPDH, GPD1, GSK3b and HK1.

**Figure supplement 1—source data 2.** Raw unedited blot images and uncropped blot images of LDHA, PDHA1, PDK1, PGD, PIK3CA and PTEN.

**Figure supplement 2.** Detailed schematic malate-aspartate shuttle based on signals derived from bioinformatic analyses of the transcriptome and proteome and on selected WBs.

**Figure supplement 3.** Representative examples of (**A**) ACACB immunolabeling of TG[AC8] and WT LV myocytes (scale bar 10 μm); (**B**) average ACACB fluorescence in LV cardiomyocytes (n=25 for each group); (**C**) Relative Quantification of Acaca mRNA expression in LV tissue (n=4 for each group).

note, GLUT1, the embryonic type of glucose transporter, *Fajardo et al., 2021*; *Smoak and Branch, 2000* and not the canonical heart form, GLUT4 (AKA Slc2a4), which was observed in the adult heart in response to myocardial stress (*Brosius, III et al., 1997*; *Razeghi et al., 2001*). Interestingly, bioinformatic analyses suggested that, both transcript (down by 86%) and protein (down by 38%) expression of ALDOB were reduced in TG[AC8] vs WT. Signals related to reduced fatty acid oxidation (FAO) likely stem from reductions in CPT1, CPT2, ACAT1, HADH, HADHA, and HADHB (*Figure 12*). Further, signals from bioinformatic analyses pointed to reduced utilization of the TCA cycle and mitochondrial respiration within the TG[AC8] LV vs WT (*Figure 12*). Interestingly, omics signals suggested that the malate-aspartate shuttle (MAS), which results in the net transport H+from cytosol back to the mitochondria, may be suppressed in TG[AC8] vs WT, because transcripts of *Ldhb, Mpc1, Mpc2, Mdh1, Got1, Slc25a11 Slc25a13*, and SLC25A13 protein was also downregulated (*Figure 12*, *Figure 12— figure supplement 2*). Reduced shuttling of aspartate into mitochondria in conjunction with increased

level of ACC2 (*Figure 12—figure supplement 3A, B*) would favor increased nucleotide synthesis (*Matsuura et al., 2020*; *Ritterhoff et al., 2020*).

The results of bioinformatic analyses also pointed to increased expression, in TG^AC8 vs WT, of the glucose transporter, lactic acid dehydrogenase type A and the glutamine transporter, that are required for enhanced aerobic glycolysis (*Vander Heiden et al., 2009*; *Figure 12*). HIF1α and other PI3K/AKT driven signaling processes in TG^AC8 suggested by omics analyses (*Figure 13*) and validated by WB (*Figure 13—figure supplement 1*) are tightly linked to aerobic glycolysis and utilization of pentose phosphate shunt (PPS) (*Vander Heiden et al., 2009*). Increased utilization of the PPS in TG^AC8 vs WT is suggested by increases in both the expression of transcripts and proteins of G6PDH, PGD, and PRPS2 (*Figure 12*).

Because Acetyl-CoA carboxylase (ACC), a complex multifunctional enzyme system that catalyzes the carboxylation of acetyl-CoA to malonyl-CoA, which limits CPT1 transport of FA into the mitochondria, we inquired whether, in addition to reduction in CPT1 transcripts and reductions in CPT1 and CPT2 proteins in TG^AC8 vs WT, ACC2 might also be increased in TG^AC8 LV myocytes. Indeed, immunolabeling of isolated LV cardiomyocytes indicated a clear increased in ACC2 (AKA ACACB) protein expression (*Figure 12—figure supplement 3A*); and an increase *Acc1* (AKA *Acaca*) mRNA was documented in LV tissue by RT-qPCR (*Figure 12—figure supplement 3B*). Such an increase of ACC2 expression in TG^AC8 may not only explain a shift from FAO to glucose utilization within the TG^AC8 LV but may also explain increased utilization of Aerobic glycolysis in TG^AC8 vs WT, which is linked enhanced utilization of PPS to increase anabolic processes such as nucleotide synthesis (*Figure 12*). Because it is known that *Acc2* deletion increases FAO and suppresses glucose utilization in the context of LV pressure overload *Matsuura et al., 2020*; *Ritterhoff et al., 2020*, it is important to recall that cardiac myocytes within TG^AC8 LV are smaller in size, and not enlarged as those in pathologic cardiac hypertrophy in response to LV pressure overload, *Dorn et al., 2003* and in which ACC2 promotes glucose utilization and reduces FAO (*Matsuura et al., 2020*; *Ritterhoff et al., 2020*). In other terms, an increase in ACC2 appears to promote enhanced glucose utilization, enhanced aerobic glycolysis, enhanced utilization of the PPS, and reduced FAO, and may be involved in the reduction of average LV cardiac myocyte size in TG^AC8 and the absence of markers of pathologic hypertrophy within the TG^AC8 LV.

## Discussion

Our results indicate that the marked increases in AC8 protein level and activity, and in PKA expression and activity in TG^AC8 directly or indirectly lead to the genotypic differences in LV structure and performance, which are rooted in a pattern of concurrently enriched or activated signaling pathways. *Figure 13* depicts a scheme, based upon our results, of physiologic performance and protection circuits that appeared to be concurrently, engaged within the TG^AC8 LV. Specifically, the circuits depicted in *Figure 13* are based upon genotypic differences in: defined LV phenotypic characteristics, signals derived from genotypic differences in transcriptome and proteome and IPA analyses, selected WB, RT-qPCR and immunolabeling analyses (*Figures 1–10*, and *Supplementary file 1a–1l*), performed having visualized the consilience of enriched or activated pathways depicted within the circuit schematic in *Figure 13*. Because many of the perspectives depicted in the concentric circuitry in *Figure 13* were derived from the results of experiments using tissue lysates, it is not implied that all aspects of the circuitry pertain to a given cell type within the heart. To facilitate the understanding of nuances depicted within the circuitry, selected features are described in more detail below.

### Cardiac structure and contractile performance circuitry

#### LV structure

Although, neither LV mass assessed via echocardiograms, nor post-mortem LV weight, differed from WT, TG^AC8 LV wall thickness was increased in TG^AC8. Because the thicker LV walls encompassed a LV cavity size that was markedly reduced at both end-diastole and end-systole compared to WT, the LV mass did not differ by genotype. Thus, because left ventricular hypertrophy is strictly defined as an increase in LV mass, *Dorn, 2007*; *Maillet et al., 2013* the TG^AC8 LV is not technically hypertrophied. Furthermore, pathologic hypertrophy markers were not increased in TG^AC8. The lack of an increase in LV mass in TG^AC8 may be attributable, in part, to reduced LV wall stress (diastolic stretch), due to

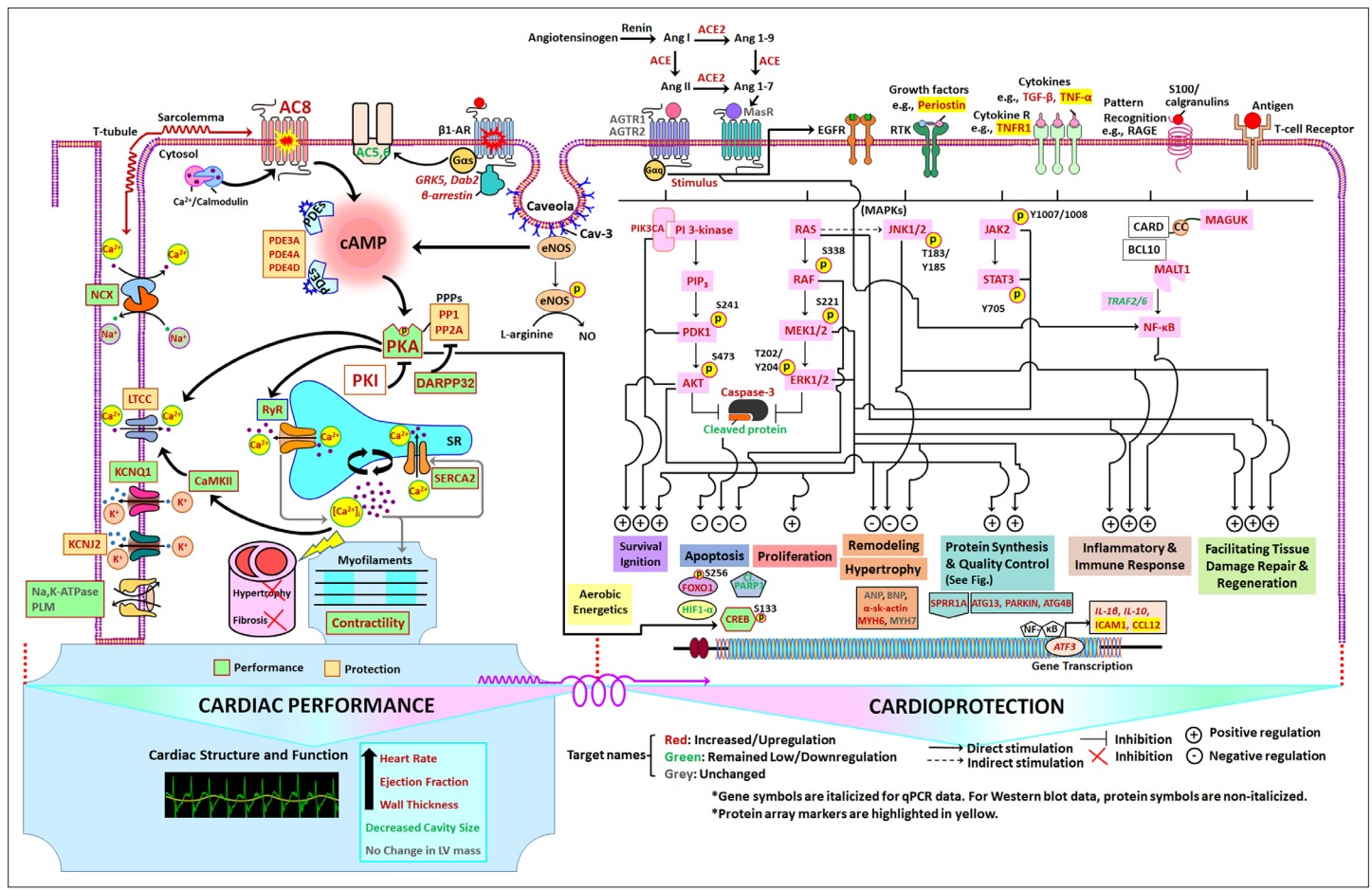

**Figure 13.** Schematic of TG[AC8] heart performance and protection circuits that appeared to be concurrently engaged in the TG[AC8] LV. The pathways/specific targets and the effector functions/outcomes culminating from the regulation of the components within the circuits in the pathways are represented according to published literature with respect to cardiac-specific context (see Discussion). Pink colors represent proteins that differed by genotype in WB. Molecular targets or components in red, green, and grey represent molecular targets or components that are increased or upregulated, decreased or downregulated, or unchanged, respectively, in TG[AC8] vs WT.

The online version of this article includes the following source data and figure supplement(s) for figure 13:

**Figure supplement 1.** WB analyses of selected proteins within the performance and protection circuitry depicted in *Figure 13*.

**Figure supplement 1—source data 1.** Raw unedited blot images and uncropped blot images of PDE4B, PDE4D, PP1, PP2A, PP2B.

**Figure supplement 1—source data 2.** Raw unedited blot images and uncropped blot images of PP2B, DARPP-32, PDE3A, PDE4A.

**Figure supplement 2.** WB analysis of selected proteins involved in Jak/Stat/Jnk/Caspase signaling.

**Figure supplement 2—source data 1.** Raw unedited blot images and uncropped blot images of Caspase3, Cleaved Caspase3, Cleaved PARP, pFoxO1 (Ser256), GAPDH, HIF1α, JAK2 (Tyr1007/Tyr1008), pCREB (Ser133), pJNK (Thr183/Tyr185), pSTAT3 (Tyr705), total CREB, total FoxO1, total JAK2, total JNK, and total STAT3.

**Figure supplement 3.** WB analysis of selected proteins involved in angiotensin receptor signaling Data are presented as Mean± SEM.

**Figure supplement 3—source data 1.** Raw unedited blot images and uncropped blot images of ACE2, AGTR1, AGTR2, and MAS1.

**Figure supplement 4.** WB analysis of Calnexin and Calreticulin, proteins involved in ER protein processing Data are presented as Mean± SEM.

**Figure supplement 4—source data 1.** Raw unedited blot images and uncropped blot images of CALR and CNX.

**Figure supplement 5.** RT-qPCR analysis of genes regulating cytokines level in the LV.

**Figure supplement 6.** Cytokine levels measured from heart tissue lysates.

**Figure supplement 7.** LV tissue staining for apoptosis and glycogen.

**Figure supplement 8.** Periostin levels detected in TG[AC8] vs WT LV (growth factor quantibody array).

**Figure supplement 9.** SPRR1A signaling network and Immunolabeling of SPRR1A.

reduced filling volume, in TG[AC8] vs WT, linked to the reduction in diastolic filling time, based on the increased heart rate. Of note in this regard, the left atrial size was not enlarged in TG[AC8] vs WT. It's of interest, however, that knock out of Protein Tyrosine Phosphatase Non-Receptor Type 1 (PTP1B aka PTPN1), which protects against pathologic hypertrophy and increases fetal gene markers in response to transverse aortic constriction, *Gogiraju et al., 2016* was increased in both RNSEQ and proteome (*Supplementary file 1c and e*).

The thickened TG[AC8] LV posterior wall revealed a cardiac myocyte profile that was shifted to cells that were smaller in size than WT. Thus, an increased density (per unit area) of smaller cardiac myocytes contributed to the increase in LV posterior wall thickness in TG[AC8]. The LV wall collagen fraction did not differ by genotype, even though TGF β protein, TGF β receptor, and downstream signaling molecules were all upregulated, suggesting that additional adaptations that uncouple TGF β signaling from its effect, including increased fibrosis, may become activated at this age in TG[AC8] LV.

## LV function

Neither the early nor the late diastolic filling rates, or early/late ratio differed by genotype, indicating that the smaller LV EDV in TG[AC8] vs WT was not associated with reduced diastolic functional measures. LV EF was markedly increased in TG[AC8] vs WT, as previously reported (*Mougenot et al., 2019*). Because stroke volume did not differ by genotype, a 30% increase cardiac output in TG[AC8] vs WT is attributable to the 30% increase in heart rate.

It bears emphasis that the resting cardiac structural and performance profiles of the TG[AC8] LV, therefore, differ from those observed in LV remodeling induced by chronic exercise endurance training, in which the diastolic cavity size, and stroke volume are increased at rest, *Fleg et al., 1994* resting HR is reduced and cardiac output is unchanged. The cardiac myocyte profile in TG[AC8] also differs from that induced in chronic exercise, in which the LV myocyte size increases, but the myocyte number is unchanged (*Moore et al., 1993*). The marked differences in cardiac structure and function between TG[AC8] and the endurance-trained heart may be attributable, in large part, to increased vagal tone at rest in the latter, but not in the former, *Moen et al., 2019* in which an exercise functional profile is maintained around the clock at rest. In other terms, the TG[AC8] heart does not have an opportunity to rest between bouts of dynamic exercise as do the hearts of endurance trained organisms.

## cAMP/PKA signaling

As expected, expression of proteins that underlie numerous, well-characterized cellular mechanisms driven by cAMP/PKA signaling that determine cardiac myocyte contractile performance and action potential characteristics (among these KCNJ2 and KCNQ1), were also upregulated in TG[AC8] vs WT in the transcriptome, proteome, both transcriptome and proteome, or in WB.

## Cardiac protection circuitry

The main *functional* category of genotypic changes identified in PROTEOMAP analysis, Environmental Information Processing (*Figure 13*), points to an integration ('consilience') of stress response pathways that were enriched and activated in TG[AC8] within the protection circuitry in *Figure 13* including: cell survival initiation, protection from apoptosis, proliferation, prevention of cardiac-myocyte hypertrophy, increased protein synthesis and quality control, increased inflammatory and immune responses, facilitation of tissue damage repair and regeneration and increased aerobic energetics.

It is noteworthy, that, in the context of the marked increased AC/PKA signaling, numerous molecules that *inhibit* AC/PKA signaling were upregulated in TG[AC8], or those that activate AC/PKA signaling are downregulated. Upstream of AC/PKA signaling numerous facets desensitization of βAR signaling (e.g. Grk5, Dab2 and β-arrestin in RNASEQ and proteome) were upregulated in the TG[AC8] vs WT LV, suggesting that signaling via βAR is downregulated in TG[AC8] vs WT. This is consistent with the results of prior studies, indicating that βAR stimulation-induced contractile and HR responses are blunted in TG[AC8] vs WT (*Moen et al., 2019*; *Liu et al., 2020*). The heart, itself, produces catecholamines and it is interesting to note that βARs become desensitized in TG[AC8], even though neither plasma norepinephrine, nor epinephrine are increased, but reduced in TG[AC8] vs WT (*Moen et al., 2019*).

Prior studies have indicated that βAR stimulation increases PI3K activity, (*Morisco et al., 2000*; *Schlüter et al., 1998*) and that PI3K signaling is involved in βAR internalization (*Nienaber et al., 2003*; *Naga Prasad et al., 2002*). Both RNASEQ and proteome signaled that PI3Kγ (*Pi3kcg*) expression is

activated in TG<sup>AC8</sup> vs WT and RT-qPCR assay detected a significant increase of the catalytic subunit (*Supplementary file 1I*). PI3K/AKT signaling is also involved in cAMP metabolism, acting as an essential component of a complex controlling PDE3B phosphodiesterase-mediated cAMP destruction (*Patrucco et al., 2004*; *Perino et al., 2006*). A blunted response to βAR stimulation has been linked to a smaller increase in L-type Ca²⁺ channel current in response to βAR stimulation in the context of increased PDE activity (*Georget et al., 2002*; *Georget et al., 2003*). WB analyses (*Figure 13—figure supplement 1A*) showed that PDE3A and PDE4A expression increased in TG<sup>AC8</sup> vs WT, whereas PDE4B and PDE4D did not statistically differ by genotype.

In addition to evidence suggesting that cardiac βAR become desensitized in TG<sup>AC8</sup> LV, and that PDE's are increased, other mechanisms that limit AC-PKA signaling were increased in TG<sup>AC8</sup> vs WT. The expression levels of PKI-inhibitor protein (PKIA), which limits signaling of downstream of PKA and of protein phosphatase 1 (PP1), were increased TG<sup>AC8</sup> vs WT in proteome and transcriptome, and in WB, respectively. The Dopamine-DARPP-32 feedback on cAMP signaling pathway was both enriched and activated in TG<sup>AC8</sup> vs WT, the LV levels of dopamine were increased, and DARPP-32 protein was increased in WB (by over twofold, *Figure 13—figure supplement 1A*). Of note, a prior study showed that plasma dopamine levels were also increased in TG<sup>AC8</sup> vs WT (*Moen et al., 2019*).

Beyond the adaptations to limit the degree to which cAMP/PKA signaling is activated, numerous canonical stress-response signaling circuits downstream of AC signaling were concurrently enriched in TG<sup>AC8</sup>, including: RTKs (PI3K/AKT, ERK-MAPK); cytokine receptors (JNK1/2, JAK2/STAT3), pattern recognition receptor RAGE (S100/calgranulins) and T-cell receptor (NF-κB) (*Figure 13*, *Figure 13—figure supplement 2*).

Pathways downstream of receptor tyrosine kinases and G-protein-coupled receptors transmit signals to promote the activation of PI3 kinase and RAS-RAF-MEK1/2-ERK1/2 (*Figure 13*). The tyrosine receptor coupled PIK3CA, AKT and PDK1 were increased in TG<sup>AC8</sup> vs WT (proteome and WB), and PTEN (WB) and PIP3 (ELISA) were also increased in TG<sup>AC8</sup> vs WT. Inactivation of GSK3B via its phosphorylation by AKT1 is a key conversion point of numerous processes that confer cardioprotection (*Juhaszova et al., 2004*). Although WB analysis indicated that GSK3B phosphorylation did not differ between genotypes, total GSK3B protein was significantly increased in TG<sup>AC8</sup> vs WT (*Figure 13—figure supplement 1A*). Receptor activation by adrenergic agonists and growth factors (such as periostin, increased in omics and validated by antibody array *Supplementary file 1e*, *Figure 13—figure supplement 8*) bind to β-AR and RTK, respectively. Ras-p21 functions as a molecular link for membrane GPCR and RTK to transduce signals from these receptors to downstream signaling machinery (*Figure 13*, ERK arm). Ras-Raf-MEK-ERK pathway plays a key role in cardioprotection in the context of ischemia-reperfusion injury and oxidative stress (*Lips et al., 2004*). Downstream of Ras-p21, c-Raf is subsequently induced, which, in turn, sequentially activates MEK1/2 and ERK1/2 to achieve their downstream effects (*Figure 13*, ERK arm). MEK/ERK1/2 activation promotes tissue repair that is essential for repair of damaged cells or cell regeneration in response to cardiac stress in vivo (*Figure 13*, ERK arm), especially those observed in genetically engineered animal models (*Bassat et al., 2017*).

Ras-p21 signaling also exerts protective influences on cell reparative and regenerative capacity, presumably via activation of ERK-independent downstream pathways of PI3K/AKT signaling *de Jonge et al., 2006*, *Rose et al., 2010*. Increased expression of the catalytic subunit PIK3CA in TG<sup>AC8</sup> may be linked to the activation of downstream targets, including PDK1 phosphorylation on serine 241, subsequent phosphorylation of AKT2 (i.e. threonine 308 and serine 473) and of the transcription factor, Foxo1, at serine 256 (*Figure 13*, PI3K arm, bottom panel). Activation of AKT inhibits hypertrophic signaling in adult hearts in vivo, *de Bosch et al., 2006*; *Antos et al., 2002*; *Moc et al., 2015*; *Maillet et al., 2013* while inducing protein synthesis and activating protein quality control signaling (*Freed et al., 2003*). This anti-hypertrophic effect of AKT in TG<sup>AC8</sup> may be linked to the lack of increased myocyte size and absence of increased LV mass in TG<sup>AC8</sup>.

The tumor suppressor PTEN is a lipid phosphatase that regulates cell growth, survival, and migration by catalyzing the dephosphorylation of the phospholipid phosphatidylinositol (3,4,5)-trisphosphate PtdIns (3,4,5)P3 or PIP3, an integral second messenger molecule for the PI3K-AKT signaling, thus antagonizing this pathway. As noted, PTEN protein abundance was increased in the TG<sup>AC8</sup> LV compared to WT. Phosphorylation of sites within the PTEN C-terminal domain, including S380, T382, and T383, have been demonstrated to be involved in the regulation of stability and activity, with

a loss in phosphorylation leading to membrane recruitment and greater activity followed by rapid proteosome-mediated degradation (*Das et al., 2003*; *Torres and Pulido, 2001*; *Vazquez et al., 2000*). Phosphorylation of Ser380 residues was significantly lower in TG$^{AC8}$ vs WT as was the ratio of p-PTEN Ser380/Total PTEN (*Figure 12—figure supplement 1*), suggesting that PTEN activity is increased in the TG$^{AC8}$ vs WT. The 2-fold higher expression of PIP3 in TG$^{AC8}$ vs WT, along with increases in PI3K catalytic subunit abundance and AKT activity, strongly suggest that PTEN does not exert **a net** negative regulatory effect on this pathway in TG$^{AC8}$. Rather, an increase in PTEN in TG$^{AC8}$ may be necessary to maintain the available pool of PIP2 that is required for phospholipid metabolism and other signaling pathways, such as PLC/IP3/DAG signaling (important in Ca$^{2+}$ regulation), or in functional processes, like Endocytosis and Actin cytoskeleton remodeling, *Czech, 2000* that are enriched and activated in TG$^{AC8}$ vs WT.

Integral components of the PI3K-AKT and RAS-RAF-MEK1/2-ERK1/2 survival-associated pathways converge to suppress apoptosis (*Yang et al., 2016*; *Abeyrathna and Su, 2015*; *Ghigo and Li, 2015*; *Figure 13*, PI3K and ERK arms). Caspase-3 levels were increased. The barely detectable or low protein expression of hallmarks of apoptosis in TG$^{AC8}$ LV, including *cleaved* caspase-3 and its downstream *cleaved* PARPs (*Figure 13* (PI3K and ERK arms top and bottom panels), *Figure 13—figure supplement 1B*), suggest that apoptosis suppression signaling mechanisms become activated in the TG$^{AC8}$ LV. That full-length caspase-3 expression was increased in TG$^{AC8}$ vs WT, but not cleaved, *Khalil et al., 2014* provides strong evidence for the activation of anti-apoptotic functions of caspase-3 in the TG$^{AC8}$ LV. Further, there was no indication of increased apoptotic nuclei in TG$^{AC8}$ LV tissue (*Figure 13—figure supplement 3A*).

PKA signaling links many pathways involved in protein synthesis, cell cycle, proliferation, and growth and developmental circuits. Regulation of the synthesis of numerous proteins related to cell cycle, cell proliferation, and growth and development were increased in TG$^{AC8}$ (*Supplementary file 1m*). For example, expression of RAS transcripts and protein were increased in TG$^{AC8}$ vs WT, and numerous molecules downstream of RAS were increased and activated in TG$^{AC8}$ LV including: RAF, MEK1/2 and ERK1/2, indicating that MEK/ERK signaling, known to be associated with cell proliferation (*Rose et al., 2010*), is activated to a greater extent in TG$^{AC8}$ vs WT (*Figure 13*, ERK arm).

AKT2 protein expression level and phosphorylation state were markedly increased in TG$^{AC8}$ vs WT, and numerous growth factor signaling molecules upstream of AKT were also upregulated in TG$^{AC8}$ vs WT (*Figure 13*) including periostin, which plays a role in tissue regeneration, including wound healing, and ventricular remodeling following myocardial infarction (*Dixon et al., 2019*; *Stansfield et al., 2009*; *Zhao et al., 2014*) *Egfr*, *Hdgf*, and *Igf1* transcripts were also increased in TG$^{AC8}$, and IGF1R protein was also borderline increased (interestingly, some features of endurance exercise conditioning involve upregulation of IGF signaling via PIK3CA *Ikeda et al., 2009*). Other molecules related to cell cycle, cell proliferation and growth and development that were increased in TG$^{AC8}$ vs WT included: CDK2, CDK6, NOTCH1, beta1 catenin protein and PCNA, *Hand2*, *Tbx5* and *Tbx20* transcripts. Although nuclear EdU labeling was not increased in cardiac myocytes, EdU labeling of non-cardiac myocytes was substantially increased in TG$^{AC8}$ vs WT (*Figure 3*).

## Protein synthesis and repair

Sprr1a, which is linked to induction of protein synthesis (*Figure 13*, bottom panel) and is the most abundantly overexpressed transcript **and** protein in the TG$^{AC8}$ **LV**, is associated with inflammation, cellular stress, and repair, *Choi et al., 2020*; *Lindborg et al., 2021* and has been linked to protection against cardiomyocyte death in the setting of ischemia–reperfusion injury.*Pradervand et al., 2004*; *Shah et al., 2021* Sprr1a stimulates the expression of Rtn4 (*Figure 13—figure supplement 9*) , a member of the reticulon encoding gene family associated with endoplasmic reticulum (ER) that is involved in neuroendocrine secretion or in membrane trafficking in neuroendocrine cells, *Steiner et al., 2004*; *van de Velde et al., 1994* which was also overexpressed in TG$^{AC8}$ vs WT.

Derlin1, another protein that was also markedly overexpressed by 132% in TG$^{AC8}$ vs WT (*Supplementary file 1e*) participates in ER associated protein degradation by recognizing and selecting misfolded or unfolded proteins and translocating these from the ER lumen to the cytosol for proteasomal degradation. Calnexin and calreticulin, both members of the CALR/CN cycle that ensures the quality and folding of newly synthesized molecules within the ER, were also both increased by WB in TG$^{AC8}$ (*Figure 13—figure supplement 1D*). Numerous other molecules involved in unfolded protein

response signaling significantly differed by genotype in omics analyses: transcripts included *Atf4, Atf6, Casp3, Calr, Hsp90*; and proteins included: CASP3, CASP6, HSP90α.

Forced changes in energy metabolism downstream of markedly increased levels of AC/cAMP/PKA signaling in TG$^{AC8}$, and resultant adaptations to this stress (*Figure 13*), are likely to produce excess wear-and-tear damage in cellular components, requiring an adaptively higher autophagy/mitophagy machinery. Several lines of evidence indicate that compared to WT autophagy was indeed increased in TG$^{AC8}$ LV. The preserved level of mitochondrial fitness (*Figure 6I and J*; *Figure 7M and N*) in TG$^{AC8}$, in the context of marked chronic AC-driven cellular stress, likely reflects one such autophagic adaptation. Many upregulated transcripts and proteins in TG$^{AC8}$ vs WT, for example, serine/threonine-protein kinases MSK1 (involved in regulation of the transcription factors CREB1, ATF1, RELA and STAT3) and MNK1 (involved in the initiation of protein translation by interacting with eIF4G and phosphorylating eIF4E), are also linked to autophagy.

## Shifts in metabolism

A greater **rate** of energy production in TG$^{AC8}$ vs WT is likely required to support not only the higher HR and increased contractility during each heartbeat compared to WT, but also increased growth factor and other PI3K-directed signaling pathways, for example autophagy, protein synthesis and protein quality control. The increased 'energy bills' of the TG$^{AC8}$ LV are apparently being 'paid in full' because there is no indication that the TG$^{AC8}$ LV is chronically hypoxic, as HIF-1α transcripts are down regulated (*Supplementary file 1b*) and HIF-1α protein level is reduced (*Figure 13—figure supplement 1B*). Further, steady levels of ATP and phosphocreatine in the WT LV in vivo at rest did not differ in TG$^{AC8}$ vs WT. Surprisingly, exercise capacity of TG$^{AC8}$ exceeds that of WT (*Esposito et al., 2008*).

Although metabolic flux was not measured in our study, bioinformatic signals (GLUT1 (aka Slc2a1) HK1 and PFK) suggested that catabolism of glucose is markedly increased in TG$^{AC8}$ vs WT (*Figure 12*). Gys1 and phosphorylase kinase (PhK) were increased in both the transcriptome and proteome in TG$^{AC8}$ vs WT, which, in the context of an increase in glycogen staining in TGAC8 vs WT, suggests that a high turnover (increased synthesis and increased degradation) of glycogen may be involved in the marked increase in glucose catabolism within the in TG$^{AC8}$ LV (*Figure 12*).

Growth factor signaling and other PI3K/AKT-driven signaling processes that are increased in TG$^{AC8}$ vs WT are tightly linked to **aerobic glycolysis,** *Vander Heiden et al., 2009*; *Wang et al., 2017* and it is well established that the **rate** at which ATP is generated in aerobic glycolysis is *greater* than that generated in oxidative phosphorylation via the TCA cycle and respiration (*Vander Heiden et al., 2009*). Bioinformatic analyses also indicated that in addition to increased expression of the glucose transporters, lactic acid dehydrogenase type A (cytosolic isoform of LDH) and the glutamine transporter, which are required for **aerobic glycolysis** *Vander Heiden et al., 2009* are also increased in TG$^{AC8}$ vs WT, suggesting that, relative to WT, the TG$^{AC8}$ heart may utilize **aerobic glycolysis** to fulfill part of its energy needs.

Aerobic glycolysis promotes utilization of PPS, which when activated, facilitates amino acid and nucleotide synthesis. Overexpression in TG$^{AC8}$ LV of genes and proteins that are operative within the glucose metabolic pathway, for example G6PDH, PGD, and PRPS2 (*Figure 12*), suggests that the PPS is more highly utilized in the TG$^{AC8}$ vs WT LV. Enhanced utilization of PPS in TG$^{AC8}$ may be involved in re-synthesis of crucial amino acids and nucleotides following their degradation (via increased autophagy, mitophagy and proteosome activity in TG$^{AC8}$ vs WT). PPS and may also be involved in catalyzing the replenishment of NADPH in TG$^{AC8}$ via LDH and increased G6PDH and PGD (*Figure 12*). Further, ACLY, which catalyzes the **synthesis** of fatty acids (FA) leading to the formation of AcCoA, linking to synthesis of FA, lipids, phospholipids, and ceramides, was increased in the proteome of TG$^{AC8}$ vs WT (*Figure 12*, *Figure 10—figure supplement 5*, *Supplementary file 1e*). Enhancement of these processes in TG$^{AC8}$ via increased utilization of PPS may be linked to signals pointing to increased cell cycle, cell proliferation and growth and development in TG$^{AC8}$ vs WT. In this context, increased utilization of aerobic glycolysis and PPS in TG$^{AC8}$ would resemble the developmental stage of **embryonic** heart differentiation and growth prior to the onset of the fetal stage, during which time respiratory enzymes, e.g., COX7 become induced (*West et al., 2018*). Thus, increased utilization of aerobic glycolysis and PPS, in TG$^{AC8}$ vs WT may be a crucial factor that underlies increased cardiac biomass (thicker LV walls vs WT containing an increased number of small myocytes, and an increased number of EdU-labeled non-cardiac myocyte nuclei, in the absence of increased collagen deposition).

In addition to pointing to increased glycolysis in TG[AC8] vs WT, bioinformatic signals suggested that the fatty acid β oxidation pathway and pathways that degrade branched chained amino acids are concurrently reduced (*Figure 12*). Bioinformatic analyses also suggested that utilization of TCA cycle and mitochondrial respiration within the TG[AC8] LV are reduced compared to WT, even though the mitochondrial number and volume do not differ by genotype.

Downregulation of transcripts or proteins operative within the MAS, suggested that in conjunction with reduced TCA cycle and mitochondrial respiration, the translocation of H+from cytosol to mitochondria to restore the level of mitochondrial NADH is also reduced in TG[AC8] vs WT. A proposed interaction between cytosolic lactate and the MAS is as follows: cytosolic lactate formed during glycolysis may be used as a mitochondrial energy source via translocation of lactate via lactate-MAS within the same cell; lactate can also be exported from one cell and taken up by other cells; *Brooks, 1998* in heart, lactate can also regenerate cytosolic NAD$^+$ from cytosolic LDH; *Safer et al., 1971*; *Kane, 2014* and under conditions when shuttling of NADH from cytosol into the mitochondria via the MAS is limited, an increase in cytosolic lactate concentration might be expected to favor cytosolic NAD$^+$ regeneration via the LDH reaction. Thus, it might be expected that the persistent high cardiac workloads that are characteristic of TG[AC8] heart, in conjunction with reduced MAS and reduced expression of proteins involved in TCA cycle and mitochondrial respiration in TG[AC8] vs WT, are associated with increased replenishment of cytosolic NADH from NAD+, at the expense of replenishment of mitochondrial NADH from NAD$^+$ within mitochondria. It is also important to note that both transcripts and protein levels of IDH, which catalyze the oxidative decarboxylation of isocitrate to α-ketoglutarate, are reduced in TG[AC8] vs WT. During a high cardiac workloads, e.g. as in the TG[AC8] heart, the α-ketoglutarate/malate transporter located within the inner mitochondrial membrane does not compete favorably with mitochondrial matrix α-ketoglutarate dehydrogenase for α-ketoglutarate their common substrate (*Kane, 2014*). It is well known that, during strenuous exercise requiring increased energy utilization, plasma lactate concentration becomes increased, even in the presence of adequate oxygen, suggesting that cytosolic lactate may be increased *O'Donnell et al., 2004* in the TG[AC8] heart, thus favoring NAD$^+$ regeneration in the cytosol (*Robergs et al., 2004*).

## ROS regulation

Although increased rates and amounts of ATP production and utilization in the TG[AC8] heart, to support simultaneous increased cardiac performance and growth factor signaling, are likely to be associated with a substantial increase in ROS generation, ROS levels were not increased in the TG[AC8] LV (*Figure 8B*). Because ATP production via aerobic glycolysis generates less ROS compared to oxidative phosphorylation via the TCA cycle and respiration, *Vander Heiden et al., 2009* increased utilization of aerobic glycolysis would be one mechanism to limit ROS production within the TG[AC8] LV. Mechanisms that reduce ROS levels that appeared to be utilized to a greater extent in TG[AC8] vs WT include NRF2 directed signaling, which was one of the top enriched and activated pathways, as well as increased SOD3 and HMOX1 transcripts and proteins (*Supplementary file 1i*) which not only protect tissue against oxidative stress, but also protect against apoptosis (*Yet et al., 2001*; *Zhao et al., 2013*). Importantly, HMOX1 is also an activator of NRF1, which has been recently discovered to be involved in heart regeneration and repair (*Cui et al., 2021*). Importantly, four members of the family of glutathione peroxidases involved in the termination reaction of ROS pathways were increased in TG[AC8] vs WT: Gpx3, Gpx7, Gpx8, Gpx1 (*Supplementary file 1e*). Several members of glutathione transferases were also differentially expressed in TG[AC8] vs WT: expression of GSTA3, GSTM5, GSTM1, and mitochondrial glutathione reductase GSR was upregulated, while expression of GSTM7 and GSTO1 was downregulated.

## Inflammation/immune signaling

The myocardium is intimately connected with the immune system and activation of the immune system has been shown to have both protective and maladaptive effects on the heart (*Adamo et al., 2020*). This inherent duality has spurred a quest for tools to harvest the protective effect of the innate and adaptive immune response without experiencing their detrimental effects (*Adamo et al., 2018*). On this background, it is remarkable that the most upregulated pathway in the TG[AC8] hearts was 'leukocyte extravasation signaling' and that the TG[AC8] was characterized by upregulation of several inflammatory molecules and pathways, both in the 'omics analysis' and in western blot based or

RT-qPCR based validation analyses. TG$^{AC8}$ hearts, in fact, showed upregulation of *IL-6*, *IL-10*, ICAM1, and CCL12 (*Figure 13—figure supplement 6*). Furthermore, TG$^{AC8}$ hearts showed upregulation of MAGUK and MALT1 of the CBM complex together with its downstream NF-κB signaling (NF-κB coactivators TRAF2/6 and ATF3 [*Figure 13*, NF-κB arm]). The TG$^{AC8}$ hearts were also characterized by phosphorylation of JNK1/2 on threonine 183 and tyrosine 185 (*Figure 13*, JNK arm) that paralleled gene expression changes consistent with activation of the JNK1/2 signaling pathway in TG$^{AC8}$ vs WT.

The fact that T cell receptor signaling and B cell receptor signaling were upregulated, together with the fact that we detected upregulation of the CBM complex (central in B and T cell activation), suggests that the AC8 heart might be characterized by recruitment of T and B cells to the myocardium (*Juilland and Thome, 2016*). Specific subclasses of lymphocytes have been shown to have protective effect on the heart (*Adamo et al., 2020*) JNK1/2 activation is centrally involved in inflammatory signaling *Craige et al., 2019* but also linked to inhibition of cardiac hypertrophy *Sadoshima et al., 2002* and promotion of reparative and regenerative capacities (*Kaiser et al., 2005*). Taken together, these observations raise the intriguing possibility that the AC8 overexpressing heart might manage to cope with continuous stress also via the activation of a *cardio-protective* immune responses.

## TG$^{AC8}$ LV protection circuits resemble some adaptive mechanisms that accompany disease states

When we attempted to ascertain the extent to which cardiac protection circuits identified in the TG$^{AC8}$ LV (*Figure 13*) might be utilized as adaptations to the stress of various disease states, the top disease categories identified in both the TG$^{AC8}$ transcriptome and proteome were **Organismal Injury and Abnormalities and Cancer**, the latter category being subsumed within the former. Because it is well known that cancers of the heart are rare, the implication of the patterns of 'cancer' in the TG$^{AC8}$ LV based on our bioinformatics analytic tools (IPA analyses) was at first puzzling to us, so we tried to rationalize what that might mean. But, as pointed out by Weinberg and Hanahan in their 'Hallmarks of Cancer' *Hanahan and Weinberg, 2000*; *Hanahan and Weinberg, 2011* mammalian cells have evolved complex concentric circuits to defend against stress that threatens their health or life, and envisioned that cancer cells successfully breach one or more of these concentrically arrayed defenses, because of somatic mutations in genes coding for proteins within this adaptive circuitry. They referred to this ability of cancer cells to breach this defense circuitry as the 'Hallmarks of Cancer', which, in reality, mirrored aspects of the mammalian complex defense circuitry that had been breached.

Thus, we envisioned that the overwhelming cardiac stress within heart cells of chronic intense AC/PKA/Ca$^{2+}$ signaling, due to the expression of a single gene, AC8, engages this conserved complex mammalian concentric defense circuitry within heart cells that is embodied in over 250 canonical signaling pathways that were differentially enriched or activated in TG$^{AC8}$ vs WT. Numerous processes within these pathways, for example, protection from apoptosis, enhanced survival, engagement of aerobic glycolysis, increased proliferation, enhanced protein synthesis and quality control, increased inflammatory and immune response, enhanced tissue damage repair and regeneration because the sequalae of marked chronically increased AC/cAMP/PKA signaling illustrated in TG$^{AC8}$ LV protection circuitry (*Figure 13*), are embodied within the '**Hallmarks of Cancer**' (*Hanahan and Weinberg, 2000*; *Hanahan and Weinberg, 2011*). We envision that engagement of this concentric protection circuitry in TG$^{AC8}$ allows its heart to cope (for at least up to a year with numerous, marked cell-tissue stressors driven by the markedly augmented AC-cAMP-PKA-Ca$^{2+}$ signaling) (*Mougenot et al., 2019*). 'Very limited versions' of this emergent cardio-protective profile in the TG$^{AC8}$ LV have been previously discovered (by reductionist biologic research) to be central to cardiac ischemic pre-conditioning *Juhaszova et al., 2005* and exercise endurance cardiac conditioning (*Vega et al., 2017*; *Fulghum and Hill, 2018*).

When our omics data reflecting engagement of concentric adaptive circuits within the TG$^{AC8}$ LV but were specifically filtered for '**cardiovascular system**' and '**cardiovascular disease**', bioinformatic analyses predicted features commonly associated with a variety of cardiomyopathies (*Supplementary file 1n and o*). The basis of this prediction, to us, seems likely to stem from the fact that omics 'insights', generated by findings in prior cross-sectional studies of experimental heart failure, may have often been misinterpreted to be causal factors of the heart failure, rather than adaptations that are activated to compensate for the causal factors of heart failure, due to a lack of longitudinal prospective.

Chronic consistent engagement of numerous adaptive pathways, however, may present a double-edged sword, compensating for the stress of chronic myocardial performance in the 'short run', but becoming maladaptive when utilized over prolonged periods of time. In fact, TG[AC8] mice have a reduced (about 30%) *Mougenot et al., 2019* median life span, and later in life (about 12 mo) marked LV fibrosis and dilated cardiomyopathy become manifest. Thus studies, conducted longitudinally over the TG[AC8] life course, are required in order to determine which of these correlated findings are cause-and-effect vs those that are simply associated. In other terms, longitudinal analysis are required to dissect out which mechanism within the 'super performing heart' of the TG[AC8] during young adulthood gradually fail over a prolonged period of time, resulting in accelerated cardiac aging and severe, dilated cardiomyopathy (*Mougenot et al., 2019*).

## Opportunities for future scientific inquiry afforded by the present results

By design, our systems approach to elucidate some general foundations of the complex adaptation circuity that protects the chronically high performing TG[AC8] heart, represents only a 'first port of call', defining some of its general features of the TG[AC8] cardiac performance/protection profile. Because many of the perspectives depicted in the scheme in *Figure 13* were derived from bioinformatic analyses of cell lysates, it is not implied that all of these circuits are present or become activated in all cell types that reside in the LV myocardium. It is apparent, however, from our results, that consilient enrichment or activation of numerous stress response pathways within the TG[AC8] LV underlie its chronic high-level performance. A similar pattern of concurrent activation of numerous concentric stress responses may be of biological significance in pathologic states, for example, to allow for proper healing in disease states, such as infarction or failure of the heart.

A most important aspect of our results, is that they identify numerous hypotheses that can be tested in future, more in depth cardiac research aimed at for example,: (1) defining the precise cardiac cell types in which protection circuitry is activated in the chronically over-worked TG[AC8] heart (leading candidates of this regenerating cell network include endothelial cells, fibroblasts, immune cells, pericytes and *most* excitedly important, telocytes; *Varga et al., 2019*) (2) providing deeper definitions of the multitude of signaling pathways that differed significantly in enrichment and activation status among cell types in TG[AC8] vs WT (*Figures 10–13*, *Supplementary file 1f and g*). This will lead to identification of novel associations among the concurrently activated adaptive mechanisms within and among cell types in TG[AC8] LV that are not intuitively linked to cardiac protection in the context of our current state of knowledge. For example, elucidation of specific details within the leukocyte extravasation pathway, the top activated IPA canonical pathway in TG[AC8], and of other highly activated canonical signaling pathways in TG[AC8] vs WT (*Figure 10C* and *Supplementary file 1f and g*), that are not usually addressed in the field of cardiac research. This will permit detection of crosstalk between cardiac myocytes and other cell types such as immune cells and others that is likely to be critical to the cardiac protective and performance-enhancing circuitry that is harbored within TG[AC8] heart (*Figure 13*). (3) Identification post-translational protein modifications e.g. via delineations of changes in the TG[AC8] LV phosphoproteome, ubiquitome, acetylome, and 14-3-3 interactome; (4) identifying specific types of interstitial cells that proliferate within the LV myocardium that was predicted by our omics results, and validated by EdU and BrdU labeling of the young adult TG[AC8] heart between 2 and 3 mo (*Figure 3*). (5) precisely defining shifts in metabolism within the cell types that comprise the TG[AC8] LV myocardium via metabolomic analyses, including fluxomics (*Cortassa et al., 2015*). It will be also important that future metabolomics studies elucidate post-translational modifications (e.g. phosphorylation, acetylation, ubiquitination and 14-3-3 binding) of specific metabolic enzymes of the TG[AC8] LV, and how these modifications affect their enzymatic activity. Finally, the results of the present study which define the consilient activation of complex concentric cardio-protective circuitry within the TG[AC8] heart at 3 mo of age, present an unprecedent research opportunity to discovery precisely how failure of complex concentric adaptive circuity leads to accelerated aging and heart failure that characterize the TG[AC8] heart later in its life (*Mougenot et al., 2019*).

# Materials and methods

## Mice

All studies were performed in accordance with the Guide for the Care and Use of Laboratory Animals published by the National Institutes of Health (NIH Publication no. 85–23, revised 1996). The experimental protocols were approved by the Animal Care and Use Committee of the National Institutes of Health (protocol #441-LCS-2019). A breeder pair of TG$^{AC8}$ over expression mice (background strain C57BL/6), generated by ligating the murine α-myosin heavy chain promoter to a cDNA coding for human TG$^{AC8}$, were a gift from Nicole Defer/Jacques Hanoune, Unite de Recherches, INSERM U- 99, Hôpital Henri Mondor, F-94010 Créteil, France (*Lipskaia et al., 2000*). Mice were bred at 3 mo and housed in a climate-controlled room with 12 hour light cycle and free access to food and water, as previously described (*Moen et al., 2019*). All assays were performed on 3-month-old males.

## Echocardiography

Mice underwent echocardiographic (Echo) examination (40-MHz transducer; Visual Sonics 3100; Fuji Film Inc, Seattle, WA) under light anesthesia with isoflurane (2% in oxygen) via nosecone, temperature was maintained at 37 °C using a heating pad. Mice were placed in the supine position; skin hair in the chest area was shaved. Standard ECG electrodes were placed on the limbs and ECG Lead II was recorded simultaneously with acquisition of echo images. Each Echo examination was completed within 10 min. Parasternal long-axis views of the LV were obtained and recorded to ensure that mitral and aortic valves and the LV apex were visualized. From the parasternal long-axis view of the LV, M-mode tracings of LV were obtained at mid-papillary muscle level. M-mode tracing of Left atrium and basal aorta were recorded at aortic valve level and Left atrial dimension (LAD) and Aortic lumen dimension (AoD) were measured. Mitral valve blood flow velocity (E and A waves) was recorded at the tip of the mitral valves at an angle of 450. Parasternal short-axis views of the LV were recorded at the mid-papillary muscle level. Endocardial area tracings, using the leading-edge method, were performed in the 2D mode (short-axis and long-axis views) from digital images captured on a cine loop to calculate the end-diastolic and end-systolic LV areas. LV End-diastolic volume (EDV) and end-systolic volume (ESV) were calculated by a Hemisphere Cylinder Model method. Stroke volume (SV) was calculated as SV = EDV ESV. Cardiac output (CO) was calculated as CO = SV*HR. Ejection Fraction (EF) was derived as EF = 100 * (EDV -ESV) / EDV. Cardiac Index (CI) was calculated as CI = CO / BW. LV Posterior Wall (PW) and Inter Ventricular Septal thicknesses (IVS) were measured from the LV M-mode tracing LV. LV mass (LVM) was calculated from EDV, IVS and PW. LV early (E) and late (A) diastolic filling rates, and early/late ratio (E/A) were calculated from mitral valve blood flow velocities.

All measurements were made by a single observer who was blinded to the identity of the tracings. All measurements were reported an average of five consecutive cardiac cycles covering at least one respiration cycle (100 times/min in average). The reproducibility of measurements was assessed by repeated measurement a week apart in randomly selected images; the repeated-measure variability was less than 5%.

Echocardiography data are expressed as mean ± SEM. Differences between two groups were assessed by a t-test. Statistical significance was assumed at *P*<0.05.

## Heart and cardiac tissue isolation

Mice were injected (intraperitoneally) with heparin and acutely anesthetized with pentobarbital-based euthanasia solution. The heart then was quickly removed and placed into cold PBS solution. The left ventricle free wall, without the septum, was identified anatomically, under a dissecting microscope, and pieces of tissue 2x3 mm were dissected and snap frozen in liquid nitrogen.

## Plasma catecholamine measurements

Plasma concentrations of catecholamines (noradrenaline, adrenaline, Dopa and dopamine) and their degradation products: 3,4-dihydroxyphenylglycol (DHPG) and 3,4-dihydroxy-phenylacetic acid (DOPAC) were quantified by high performance liquid chromatography with electrochemical detection. Concentrations of catecholamines were determined after extraction from plasma using alumina adsorption according to previously described methods (*Eisenhofer et al., 1986*).

## LV histology

The LV free wall, (excluding the septum) was cut into base, mid-portion, and apex segments, fixed with formalin, embedded in paraffin, and sectioned (5 µm in thickness). Sections were stained with hematoxylin and eosin, silver (Reticulum stain kit, American MasterTech Scientific, Inc, Lodi, CA) and Masson's trichrome (American MasterTech Scientific, Inc, Lodi, CA). Myocyte cross-sectional area was measured from images captured from silver-stained 5-µm-thick sections of the LV mid portion sections. Suitable cross sections were defined as having nearly circular-to-oval myocytes at the nuclear level. Outlines of ~25 myocytes were traced in each section. Morphometric Analyses were performed using the computerized imaging program MetaMorph (MetaMorph Imaging System, Universal Imaging Corp) using light microscopy.

## EdU labeling for detection and imaging of cardiac cell DNA synthesis

Mice were administered 0.35 mg/L 5-ethynyl-2'-deoxyuridine (EdU) for 28 days via drinking water, changed every third day. Mice were then administered pentobarbital IP, heart removed and placed in PBS. The aorta was cannulated, and the heart perfused with PBS for 5 min followed by perfusion at ~100 mmHg with 4% paraformaldehyde for approximately 10 min or until flow rate was greatly reduced. Hearts were stored in fresh 4% formaldehyde for 24 hr at 4 degrees. Hearts were washed with phosphate buffer and imbedded in 4% low melting point agarose. Sequential transverse sections from the heart of WT and TG$^{AC8}$ transgenic mice were sectioned on the Leica Vibratome VT1000s from 200 u-300uM. Sections were permeabilized with 0.2% triton, glycine, and 2% DMSO for 3 days. EDU labeling (Click Chemistry Tools), and primary antibodies Vimentin 1:500 (Synaptic Systems), Actinin (Sigma) 1:500, WGA 1:500 (Vector Labs), 4',6-diamidino-2-phenylindole DAPI 1:300 (Sigma). Five microscopic fields, per mouse, in the left ventricle were visualized in cardiomyocytes via fluorescent imaging (Zeiss LSM 980) at ×400 magnification, and the number of cardiac nuclei staining positively for EdU was counted in each.

## BRDU labeling

To monitor cardiomyocyte S-phase activity, TG$^{AC8}$ mice were crossed with MHC-nLAC mice (which express a nuclear-localized β-galactosidase reporter under the transcriptional regulation of the mouse α-cardiac MHC promoter; these mice are useful to identify cardiomyocyte nuclei in histologic sections) *Soonpaa et al., 1994*. The resulting TG$^{AC8}$, MHC-nLAC double-transgenic mice and MHC-nLAC single-transgenic mice were identified and sequestered. At 28-to-30 days of age, the mice were administered BrdU via drinking water (0.5 mg/ml, changed every 2nd day) for a total of 12 days. Hearts were then harvested, fixed (1% paraformaldehyde, 50 mM cacodylate, 0.665% NaCl, pH 7.3) for 24 hr at 4 °C, cryopreserved (30% sucrose) for 24 hr at 4 °C, embedded and cryosectioned at 10 µm using standard methods (*Junqueira and Kelley, 1992*). Sections were subjected to antigen retrieval (10 mM trisodium citrate, 0.05% Tween20, pH 6) for 30 min at 100 °C, and non-specific signal was then blocked using MOM blocking reagent (Vector Labs, Burlingame California) following manufacture's recommendations. Sections were then processed for β-galactosidase (#A-11132, Invitrogen Life Sciences, Grand Island New York) and BrdU (#11296736001, Roche, Indianapolis Indiana) immune reactivity; signal was developed using Alexa 555 goat anti-rabbit for β-galactosidase and Alexa 488 goat anti-mouse for BrdU (secondary antibodies were #A21429 and #A1100, respectively, Invitrogen). Sections were counterstained with Hoechst 33342 (Sigma-Aldrich, St. Louis Missouri, blue signal) and cover slipped. After processing, the sections were imaged sequentially for the red, green and blue signals using Surveyor software (version 9.0.4.5, Digital Imaging Systems Ltd., Buckinghamshire, UK) interfaced with a Leica DM5500 (Leica AG, Wetzlar, Germany) microscope. The percentage of S-phase cardiomyocyte nuclei (as evidenced by the overlay of red βGAL signal and green BrdU immune reactivity) was then quantitated.

## Electron microscopy

Mice left ventricles were dissected and processed for transmission electron microscopy visualization. Fixation for electron microscopy was performed using 2.5% glutaraldehyde in 0.1 M sodium cacodylate buffer, pH 7–7.4. Samples were post fixed in 1% osmium tetroxide for 1 hr at 4 °C in the same buffer, dehydrated and then embedded in Embed 812 resin (Electron Microscopy Sciences, Hatfield, PA) through a series of resin resin-propylene oxide gradients to pure resin. Blocks were formed in

fresh resin contained in silicon molds, and the resin was polymerized for 48–72 hr at 65 °C. Blocks were trimmed and sectioned in an EM UC7 ultramicrotome (Leica Microsystems, Buffalo Grove, IL) to obtain both semi-thick (0.5–1 µm width) and ultrathin (40–60 nm width) sections. Semi-thick sections were mounted on glass slides and stained with 1% toluidine blue in a 1% borax aqueous solution for 2 min. Micrographs were obtained using a Leica AXIO Imager light microscope with a Axiocam 512 color camera (Carl Zeiss, White Plains, NY). Ultrathin sections were stained with uranyl acetate and lead citrate, and then imaged on a FEI Tecnai G$^2$ 12 Transmission Electron Microscope (TEM) with a Gatan OneView 16 Megapixel Camera.

## Electron microscopy image analysis

Micrographs at ×1200 and x2900 magnification were obtained from randomly selected areas of cardiomyocytes cytoplasm for illustration and quantitative analysis purposes of mitochondrial population and lipid droplets quantification. We determined two stereological parameters: (a) Na, which is the numerical profile density (number of figures of interest / µm2 of cell fraction) and (b) volume density of figures of interest (Vv; i.e., the volume fraction of cardiomyocyte cytoplasm occupied by figures of interest). Volume density was obtained following a point analysis using a simple square lattice test system (*Weibel, 1979*). Stereological measurements were performed using ImageJ software (NIH). Mitochondria were identified as electron dense double membrane organelles vesicles with identifiable cristae. Damaged mitochondria presented swollen and disrupted electron-'lighter' cristae. Lipid droplets were denoted as electron-light, not-limited by any membrane vesicles with very clear and homogeneous content. For stereological analysis, only micrographs depicting longitudinal sections of cardiomyocytes with visible sarcomeres were utilized, and from each of these 10 pictures were taken from four to six cells/fibers. From each picture, mitochondria were counted and measured to determine the number and area, doing the average of these metrics in each picture, and finally the average for each cell. With this procedure, a total of ~500 mitochondria per animal were counted/measured.

## Adenylyl cyclase activity in cell membranes of LV tissue

Pieces of left ventricular (LV) tissues from wild type and AC8-TG mice were frozen in liquid nitrogen, homogenized with Bel-Art SP Scienceware liquid nitrogen-cooled Mini Mortar and stored at –80 °C till use in the AC assay. On the day of the assay, 1 ml of ice-cold Lysis buffer (LB) was added to each sample (LB composition: 10 mM Tris, pH 7.6, 0.5 mM DTT, 1 mM EGTA, 0.2 mM IBMX and 0.33% PIC). Samples were sonicated on ice (3x15 s at setting 2, with 15 s of rest between bursts). 1800 µl of lysates (combined from two mice) were used for membrane isolation. To each such combined sample a Sample Separation Buffer (SSB, composition: 10 mM Tris, pH 7.6, 0.5 mM DTT, 1 mM EGTA, and 0.01% PIC) was added to a total volume of 11 ml. These samples were centrifuged for 10 min at 1000xg to remove big clumps. Supernatants were further centrifuged for 30 min at 48,254xg at 40 C in the Ultra-Clear 14x89 mm ultracentrifuge tubes filled almost to the top with SSB. Membrane proteins precipitated on the bottom of the tubes were washed three times with SSB via ultracentrifugation in the same conditions. At the end of the last wash the pure pellets were resuspended in 300 µl of the Sample Reaction Buffer (SRB, composition: 70 mM Tris, pH 7.6, 0.5 mM DTT, 1 mM EGTA, 5 mM MgCl2, 0.2 mM IBMX, and 0.33% PIC) via sonication (on ice, 3 s x 10 bursts on setting 2, rest between bursts 15 s). Protein content in the membrane preparations was quantified using a Reducing Agent Compatible Pierce Microplate BCA Protein Assay Kit # 23252.

Purified LV membranes were further diluted with SRB to a 0.2 µg/µl protein concentration and used in the AC reaction. For the AC activity detection reaction, Stock AC reaction media (SACRM) was prepared: 70 mM Tris (pH 7.6), 0.5 mM DTT, 1 mM EGTA, 5 mM MgCl2, 0.2 mM IBMX, 4 mM ATP, 20 mM Creatine Phosphate, and 240 U/ml Creatine phosphokinase. The end AC reaction composition was: 70 mM Tris (pH 7.6), 0.5 mM DTT, 1 mM EGTA, 5 mM MgCl2, 0.2 mM IBMX, 0.25% PIC, 1 mM ATP, 5 mM Creatine Phosphate, 60 U/ml Creatine phosphokinase, 0.2% DMSO, and 0.15 µg/µl membrane proteins.

To proceed the reaction, to the tube, preheated to 35 °C with 25 µl of the SACRM, 75 µl of the membrane sample was added. The AC reaction lasted for 5 min at 35 °C at 400 RPM, and was stopped by immersion of the tube into a 100 °C steel shot for 5 min. Then tube was cooled down, centrifuged for 5 min at 40 °C at 15,000xg, and the supernatant was used for cAMP measurement. 0 time samples were prepared the following way: first to the tubes 75 µl of membrane proteins were added, then

proteins were denatured for 5 min at 100 °C and cooled down, after that 25 µl of the SACRM was added to the tubes, and they were immediately immersed into 100 °C steel shot for 5 min. Then these 0 time samples were processed exactly the same way as other samples.

The cAMP concentration was quantified via a LANCE kit protocol (Lance cAMP384 kit 500 points, Perkin Elmer, AD0262) in 96-well OptiPlates (Perkin Elmer). Fifteen µl of the sample was used in the LANCE assay in a total volume of 40 µl (including 5 µl of 2 x cAMP antibodies and 20 µl of the Detection Mix). 2 x cAMP antibodies stock preparation: 40 µl of Ab stock, 107 µl of 7.5% BSA, 1853 µl of Detection buffer. Preliminary experiments demonstrated that cAMP LANCE standard curves depend a lot on the buffer in which they were prepared; because of that cAMP standards were prepared in the same buffers that were used for samples (75% SRB, 25% SACRM), they were heated, cooled down and centrifuged. Right before fluorescence detection, plates were centrifuged for 2 min at 1000xg to remove bubbles from the wells and to increase accuracy of the measurements. All measurements were done in triplicate and the average of the 3 taken as the cAMP value of that sample. Three pairs of WT and three pairs of TG[AC8] mice LV samples were used in this experiment. Statistic – values as mean ± St Error.

## Immunostaining of isolated intact mice ventricular myocytes for ADCY8, SPRR1A, and ACACB detection

Immunolabeling was performed in freshly isolated LV mouse cells. Cells were plated on laminin-coated MatTek dishes for 1 hr, 4% paraformaldehyde for 10 min, washed three times with PBS, and then permeabilized with 0.2% Triton X-100 in PBS for 10 min at room temperature. The plates were washed two more times with PBS and then incubated with 10% goat serum for 1 hr to minimize nonspecific staining. Afterwards, samples were incubated at 4 °C overnight with primary antibodies against SPRR1 (ab125374), ADCY8 (bs-3925R), and ACACB(sc-390344). Cells were then washed three times with PBS and incubated with fluorescence-conjugated secondary antibodies (1:1000) (Sigma, USA) for 45 min at 37 °C. Cell nuclei were labeled with DAPI (Sigma, USA). Cells were visualized using a LSM 710 laser-scanning confocal microscope (Carl Zeiss) and images were captured using the Carl Zeiss Zen software. Quantitative fluorescence image analysis was performed with ImageJ software, according to the following protocol: http://theolb.readthedocs.io/en/latest/imaging/measuring-cell-fluorescence-using-imagej.html. Images of stained cells were transferred and analyzed with Image J software to calculate the basic characteristics of each image, including Area, Mean Gray Value and Integrated Density. To calculate the corrected total cell fluorescence (CTCF). Small areas of positively stained fluorescent cells were selected using a free hand selection tool. A background reading was created by selecting a negatively stained rectangular section near the analyzed cell. Total fluorescence per cell was calculated in Excel with the following formula:

$$\text{CTCF} = \text{Integrated Density(Area of selected cell X Mean fluorescence of background readings)}$$

## Immunostaining of isolated intact mice ventricular myocytes for RyR2 detection

Immunostaining was performed as previously described. Specifically, freshly isolated mice ventricular myocytes from TG[AC8] and WT control mice were fixed with 4% paraformaldehyde, permeabilized with 1% Triton and incubated with blocking solution (1×PBS containing 2% IgG-free BSA +5% goat serum +0.02% NaN3 +0.2% Triton). Then, the cells were incubated with primary antibody anti-total RyR (Santa Cruse, R128, 1:500) overnight. After several wash, secondary Atto 647N-conjugated anti-mouse IgG (Sigma-Aldrich, 1:500) antibody was used, and only secondary antibody was applied to negative controls, which displayed negligible fluorescence. Confocal images of middle section were obtained via Zeiss LSM 510 (Carl Zeiss Inc, Germany) using 633 nm laser to excite the fluorophore Atto 647 N. The images were analyzed using ImageJ software (1.8 V, Wayne Rasband, National Institutes of Health). Please note that the RyR2 immunolabeling in TG[AC8] mice ventricular myocytes was saturated if using the same sampling setting as WT controls. So, the sampling condition was adjusted when sampling ventricular myocytes from TG[AC8] mice, and the density was converted to the same setting as WT control for comparison (*Yang et al., 2012*).

### Protein synthesis

Protein synthesis was assessed by SUnSET-Western Blot as previously described 6. Briefly, the puromycin solution was prepared in PBS, sterilized by filtration, and a volume of 200 µl was injected in

mice intraperitoneally, to achieve a final concentration of 0.04 μmol/g of body mass. After 30 min, mice were sacrificed, the LV was harvested and snap frozen in liquid nitrogen. Protein extraction was performed using Precellys, quantified with BCA assay 25 μg of total protein were separated by SDS-PAGE; proteins were then transferred onto PVDF membrane and incubated overnight in the anti-puromycin primary antibody (MABE343, Sigma-Aldrich, St. Louis, MO). Visualization of puromycin-labeled bands was obtained using horseradish peroxidase conjugated anti-mouse IgG Fc 2 a secondary antibody (Jackson ImmunoResearch Laboratories Inc, West Grove, PA, USA), using Pierce Super Signal ECL substrate kit (Pierce/ Thermo Scientific Rockford, IL). Chemiluminescence was captured with the Imager AI600 and densitometry analysis was performed using ImageQuantTL software (both by GE, Boston, MA). Total protein was used as control for protein loading.

Genotypic differences of protein synthesis were tested via an as unpaired t-test.

## Proteosome activity assay

Flash frozen tissue was homogenized in ice-cold cytosolic extraction buffer (50 mM Tris-HCl pH 7.5, 250 mM Sucrose, 5 mM MgCl$_2$, 0.5 mM EDTA, and 1 mM DTT). A bicinchoninic acid (BCA) assay (Pierce) was used to determine the protein concentrations. All samples were equally concentrated in proteasome assay buffer (50 mM Tris-HCl pH 7.5, 40 mM KCl, 5 mM MgCl$_2$, and 1 mM DTT). Proteasome activity was determined in the presence of 28 μM ATP using the Suc-LLVY-AMC (18 μM, Boston Biochem #S280) fluorogenic substrate with and without proteasome inhibition (MG 132, 1 mM, Sigma). The plate was read at an excitation wavelength of 380 nm and an emission wavelength of 469 nm using a Spectramax M5 (Molecular Devices). Activity was calculated by subtracting the background (proteasome inhibited value) from the reading (proteasome activated value).

## Protein aggregation assays

Protein aggregates were measured using Proteostat (Enzo, ENZ-51023) following the manufacturer's instructions. For this assay left ventricle myocardial lysate (Cell Signaling lysis buffer) was obtained, protein concentration assayed (BCA assay [Pierce]). Ten μg of protein loaded into a 96-well microplate and protein aggregates were analyzed using the Proteostat assay kit (Enzo Life Sciences) following the manufacturer's instructions. Background readings were subtracted from sample recordings and were normalized to wild-type values.

## Quantibody mouse inflammation and periostin arrays

Fresh LV tissue from 3 months old mice was homogenized and lysed in RIPA buffer (Sigma Aldrich) supplemented with protease inhibitor (Roche Inc) using a Precellys homogenizer with the CKMix Tissue Homogenizing Kit. The supernatant was collected after centrifugation at 10,000×g for 10 min at 4 °C. The protein concentration was determined using the Bicinchoninic Acid (BCA) Assay (Thermo Fisher Scientific). The assay was performed using the Quantibody Mouse Inflammation Array kit (QAM-INF-1–1, RayBiotech Inc). The samples (tissue lysates) were diluted 2 x using the sample diluent. 100 μl of both samples and standards were loaded on to the glass slide (labelled with the 40 different cytokines and chemokines) and incubated for 2 hr at room temperature. Samples and standards were decanted and from each well and washed with 150 μl of 1 X Wash Buffer I at room temperature. The detection antibody cocktail was added to each well and incubated at room temperature for 1 hr. The samples were decantyed and washed with 150 μl of 1 X Wash Buffer I at room temperature. A Cy3 equivalent dye-conjugated streptavidin was added to each well and incubated in dark at room temperature for 1 hr. The samples were decanted from each well, washed with 1 x wash buffer and dried. The slide was visualised for signals using a laser scanner equipped with a Cy3 wavelength (green channel). The data was extracted, computed in the standard format as provided by the company and analyzed for relative levels of different cytokines in both WT and TG$^{AC8}$ mice. Periostin was assessed in an array-based ELISA system (Growth Factor Quantibody Array, RayBiotech Life, Inc, Peachtree Corners, GA). Briefly, after a blocking step, 100 μL of LV-tissue lysates were added on a slide containing a periostin antibody, and incubated 4 °C overnight. Nonspecific proteins were then washed off, and the arrays incubated with a cocktail of biotinylated detection antibodies, followed by a streptavidin-conjugated fluorophore. Signals were visualized using a fluorescence laser scanner. Relative quantification was calculated using the median, after subtracting the background intensity.

## PKA activity

Enzymatic activity assays with heart tissue lysates were performed for cAMP-dependent protein kinase (PKA) using the PKA Kinase Activity Assay Kit (Abcam, ab139435), in accordance with the manufacturer's instructions. The units are expressed in OD/mg protein/min.

## Western blotting

Snap-frozen left ventricle (LV) tissue from 3-month-old mice was homogenized and lysed in ice cold RIPA buffer (Thermo Fisher Scientific: 25 mM Tris-HCl (pH 7.6), 150 mM NaCl, 1% NP-40, 1% sodium deoxycholate, 0.1% SDS) supplemented with a Halt protease inhibitor cocktail (Thermo Fisher Scientific), Halt phosphatase inhibitor cocktail (Thermo Fisher Scientific) and 1 mM phenylmethyl sulfonyl fluoride, using a Precellys homogenizer (Bertin Instruments) with tissue homogenization kit CKMix (Bertin Instruments) at 4 °C. Extracts were then centrifuged at 10,000×g for 10 min at 4 °C and the protein concentration of the soluble fraction determined using the Bicinchoninic Acid (BCA) Assay (Thermo Fisher Scientific). Samples were denatured in Laemmli sample buffer (BioRad Laboratories) containing 355 mM 2-mercaptoethanol at 95 oC for 5 min, and proteins (10–50 µg/lane) resolved on 4–20% Criterion TGX Stain Free gels (Bio-Rad Laboratories) by SDS/PAGE. Gels then exposed to UV transillumination for 2.5 min to induce crosslinking of Stain Free gel trihalo compound with protein tryptophan residues. Proteins were then transferred to low fluorescence polyvinylidene difluoride (LF-PVDF) membranes (BioRad Laboratories) using an electrophoretic transfer cell (Mini Trans-Blot, Bio-Rad). Membrane total protein was visualized using an Amersham Imager 600 (AI600) (GE Healthcare Life Sciences) with UV transillumination to induce and a capture fluorescence signal. Blocked membranes (5% milk/tris-buffered saline with Tween-20, TBST) were incubated with the following primary antibodies: anti-MYH6 (MA5-27820) at 1:2500 working concentration, anti-ANP (PA5-29559) at 1:1000, and anti-BNP (PA5-96084) at 1:1000 from ThermoFisher Scientific; anti-MYH7 (ab173366) at 1:500, anti-α-Sk. Actin (ab179467) at 1:1000, and anti-SERCA2 ATPase (ab91032) at 1:2000 from Abcam. Primary antibodies were then detected using horseradish peroxidase (HRP) conjugated antibody (Invitrogen) at 1:10,000. All antibodies used in our study provided in *Supplementary file 1q*. Bands were visualized using Pierce SuperSignal West Pico Plus ECL substrate kits (Thermo Scientific), the signal captured using an Amersham Imager 600 (AI600) (GE Healthcare Life Sciences) and quantified using ImageQuant TL software (GE Healthcare Life Sciences). Band density was normalized to total protein.

## RT-qPCR

RT-qPCR of LV tissue was performed to determine the transcript abundance of human AC8, genes that mediate neural autonomic input to LV (n=4 WT and 4 TG[AC8] mice) and to detect genes regulating cytokines level in the heart (n=6 in WT and TG[AC8]). RNA was extracted from left ventricular myocytes (VM) with RNeasy Mini Kit (Qiagen, Valencia, CA) and DNAse on column digestion. The cDNA was prepared using MMLV reverse transcriptase (Promega). RT-qPCR was performed using a QuantStudio 6 Flex Real-Time PCR System (Thermo Fisher Scientific) with a 384-well platform. The reaction was performed with a FastStart Universal SYBR Green Master Kit with Rox (Roche) using the manufacturer's recommended conditions; the sizes of amplicons were verified. Each well contained 0.5 µl of cDNA solution and 10 µl of reaction mixture. Each sample was quadruplicated and repeated twice using de novo synthesized cDNA sets. Preliminary reactions were performed to determine the efficiency of amplification. RT-qPCR analysis was performed using the ddCt method. Primers were selected with Primer Express 3.0 software (Applied Biosystems). Full list of primers used for amplification provided in *Supplementary file 1p*.

## RNASEQ

LV RNA was extracted from 8 of TGAC8 and WT animals. Following a quality control check, RNA was processed with a SMARTer Stranded Total RNA-Seq Kit - Pico Input Mammalian (Takara Bio USA, Inc). 75 bp single end reads generated 30–40 million reads per library. Raw RNA sequencing (RNASeq) reads were aligned and after quality trimming was mapped to the UCSC mm10 mouse reference genome and cDNA of human AC8 and assembled using Tophat v2.0 to generate BAM files for each sample. Cufflinks v.2.1.1 was used to calculate FPKM (Fragments per Kilobase of transcript per Million

mapped reads) for each sample. Differential gene expression analysis was performed with a Cuffdiff package (Cufflinks v2.1.1).

## LV proteome analysis

Four LV samples from WT and TG[AC8] mouse hearts were snap frozen in liquid nitrogen and stored at –80 °C. On average, 2 mg of muscle tissue from each sample was pulverized in liquid nitrogen and mixed with a lysis buffer containing (4% SDS, 1% Triton X-114, 50 mM Tris, 150 mM NaCl, protease inhibitor cocktail (Sigma)), pH 7.6. Samples were sonicated on ice using a tip sonicator for 1 min with 3 s pulses and 15 s rest periods at 40% power. Lysates were centrifuged at +4 °C for 15 min at 14,000 rpm, aliquoted and stored at –80 °C until further processing. Protein concentration was determined using commercially available 2-D quant kit (GE Healthcare Life Sciences). Sample quality was confirmed using NuPAGE protein gels stained with fluorescent SyproRuby protein stain (Thermo Fisher).

In order to remove detergents and lipids 500 µg of muscle tissue lysate was precipitated using a methanol/chloroform extraction protocol (sample:methanol:chloroform:water – 1:4:1:3) (*Wessel and Flügge, 1984*). Proteins were resuspended in 50 µl of concentrated urea buffer (8 M Urea, 150 mM NaCl [Sigma]), reduced with 50 mM DTT for 1 hour at 36 °C and alkylated with 100 mM iodoacetamide for 1 hr at 36 °C in the dark. The concentrated urea/protein mixture was diluted 12 times with 50 mM ammonium bicarbonate buffer, and proteins were digested for 18 hr at 36 °C, using trypsin/LysC mixture (Promega) in 1:50 (w/w) enzyme to protein ratio. Protein digests were desalted on 10x4.0 mm C18 cartridge (Restek, cat# 917450210) using Agilent 1260 Bio-inert HPLC system with a fraction collector. Purified peptides were speed vacuum dried and stored at –80 °C until further processing.

A subset of 8 muscle samples (100 µg) each corresponding to 4 controls and 4 TGAC8 LVs and one averaged reference sample were labeled with 10-plex tandem mass spectrometry tags (TMT) using standard TMT labeling protocol (Thermo Fisher). 200 femtomole of bacterial beta-galactosidase digest (SCIEX) was spiked into each sample prior to TMT labeling to control for labeling efficiency and overall instrument performance. Labeled peptides from 10 different TMT channels were combined into one experiment and fractionated.

## High-pH RPLC fractionation and concatenation strategy

High-pH RPLC fractionation was performed in an Agilent 1260 bio-inert HPLC system using a 3.9 mm X 5 mm XBridge BEH Shield RP18 XP VanGuard cartridge and a 4.6 mm X 250 mm XBridge Peptide BEH C18 column (Waters). The solvent contained 10 mM ammonium formate (pH 10) as mobile phase (A), and 10 mM ammonium format and 90% ACN (pH 10) as mobile-phase B 9.

TMT labeled peptides prepared from the ventricular muscle tissues were separated using a linear organic gradient from 5% to 50% B over 100 min. Initially, 99 fractions were collected at 1 min intervals. Three individual high-pH fractions were concatenated into 33 master fractions at 33 min intervals between fractions (fraction 1, 34, 67=master fraction 1, fraction 2, 35, 68=master fraction 2 and so on). Combined fractions were speed vacuum dried, desalted and stored at –80 °C until final LC-MS/MS analysis.

## Capillary nano-LC-MS/MS analyses

Purified peptide fractions were analyzed using UltiMate 3000 Nano LC Systems coupled to the Q Executive HF Orbitrap mass spectrometer (Thermo Scientific, San Jose, CA). Each fraction was separated on a 35 cm capillary column (3 µm C18 silica, Hamilton, HxSil cat# 79139) with 200 µm ID on a linear organic gradient at a 500 nl/min flow rate. Gradient applied from 5 to 35% in 205 min. Mobile phases A and B consisted of 0.1% formic acid in water and 0.1% formic acid in acetonitrile, respectively. Tandem mass spectra were obtained using Q Exactive HF mass spectrometer with a heated capillary temperature +280 °C and spray voltage set to 2.5 kV. Full MS1 spectra were acquired from 300 to 1500 m/z at 120,000 resolution and 40ms maximum accumulation time with automatic gain control [AGC] set to $3 \times 10^6$. Dd-MS2 spectra were acquired using a dynamic m/z range with fixed first mass of 100 m/z. MS/MS spectra were resolved to 30,000 within of a maximum accumulation time, 120ms with AGC target set to $2 \times 10^5$. Twelve most abundant ions were selected for fragmentation using 28% normalized high collision energy. A dynamic exclusion time of 45 s. was used to discriminate against the previously analyzed ions.

## Bioinformatics analysis of the LV proteome

Acquired raw data files from Q Exactive HF were converted to mascot generic format (MGF) using MSConvert, an open source software developed by ProteoWizard (http://proteowizard.sourceforge. net). Conversion filters were specified as follows: at MS level 1 with activation:HCD, threshold:count 900 most-intense, zeroSamples:remove Extra 1-, peakPicking;true 1-. Produced MGF files were searched in Mascot against the SWISS-PROT mouse database (02/06/2017) with the following parameters: enzyme trypsin/P, 2 missed cleavages, MS1 tolerance 20 ppm, MS2 tolerance 0.08 Da, quantification TMT10plex. Variable modifications were set to methionine oxidation, carbamidomethylation of cysteines, deamidation at glutamine and asparagine, carbamylation of lysin. Searched mascot data files were processed using commercially available Scaffold Q+software package (Proteome Software, Inc). Files were merged in to one summary file using MudPIT algorithm, and researched against SWISS-PROT mouse database (02/06/2017), using XTandem search engine for deeper protein coverage with both protein prophet scoring algorithm and protein clustering analysis turned on. Raw reporter ion intensities from unique peptides were extracted into excel file and used in the final analysis.

Minor variations in protein amounts between TMT channels was adjusted by calculating a ratio between signal intensity in each TMT channel (In). Adjusted intensity for each channel (Incorr) was calculated by taking a sum of all intensities in each TMT channel divided by the average ($\mu$) of all calculated sum intensities and multiplied by the initial intensity in each TMT cannel for each peptide:

$$(\sum I126 + \sum I127 + \sum I128 + \sum I129 + \sum I130 + \sum I131)/6 = ; \sum In/\mu * In = Incorr$$

Fold change for each unique peptide in each experiment was calculated by dividing Incorr by the median of all intensities in all TMT channels. The fold change between genotypes for each expressed protein was calculated by taking a median of fold change for all unique peptides of a given protein detected by mass spectrometry. Genotype differences were compared via Student's t-test. A p-value <0.05 was considered to be significant.

## MR spectroscopy high-energy phosphate

In vivo MRI/MRS experiments were performed on a Bruker spectrometer equipped with a 4.7 T/40 cm Oxford magnet and actively shielded gradients. A one-dimensional 31 P chemical shift imaging (1D-CSI) sequence was used to obtain high-energy phosphate data, as previously described (*Gupta et al., 2012*). The PCr and [β-P] ATP peaks in 31 P MR localized spectra were quantified by integration of the peak areas (*Gupta et al., 2012*).

## ROS measurements

ROS measurements were conducted using electron paramagnetic resonance (EPR) spectroscopy as previously described (*Chelko et al., 2021*). Snap-frozen heart tissue (apex, ~20 mg) was homogenized in phosphate-buffered saline (PBS) containing protease inhibitor cocktail (Roche Applied Science, Indianapolis, IN) and 0.1 mM of the metal chelator, diethylenetriaminepentaacetic acid (DTPA), at pH 7.4. Nonsoluble fractions were removed by centrifugation at 15,000 g for 10 min (4 °C). The homogenates were kept on ice and analyzed immediately. Stock solutions of 1-hydroxy-3-methoxy carbonyl-2,2,5,5-tetramethyl-pyrrolidine hydrochloride (CMH; Enzo Life Sciences, Farmingdale, NY) were prepared daily in nitrogen purged 0.9% (w/v) NaCl, 25 g/L Chelex 100 (Bio-Rad) and 0.1 mM DTPA, and kept on ice. The samples were treated with 1 mM CMH at 37 oC for 2 min, transferred to 50 µl glass capillary tubes, and analyzed immediately on a Bruker E-Scan (Billerica, MA) EPR spectrometer at room temperature. Spectrometer settings were as follows: sweep width, 100 G; microwave frequency, 9.75 GHz; modulation amplitude, 1 G; conversion time, 5.12ms; receiver gain, 2x103; number of scans, 16. EPR signal intensities were normalized with respect to the protein concentrations of the tissue homogenates as determined by Pierce BCA protein assay kit (Life Technologies).

## Determination of mPTP-ROS threshold

Experiments were conducted as described previously, *Juhaszova et al., 2004* using a method to quantify the ROS susceptibility for the induction of mPTP in individual mitochondria within cardiac myocytes (*Zorov et al., 2000*). Briefly, isolated cardiomyocytes were resuspended in HEPES buffer: 137 mM NaCl, 4.9 mM KCl, 1.2 mM MgSO$_4$, 1.2 mM NaH$_2$PO$_4$, 15 mM glucose, 20 mM HEPES, and 1.0 mM CaCl$_2$ (pH to 7.3). To assess the susceptibility of the mPTP to induction by ROS, cells were

loaded with 100 nM tetramethylrhodamine methyl ester (TMRM; Invitrogen I34361) for at least 2 hr at room temperature. Cells were imaged with an LSM-510 inverted confocal microscope, using a Zeiss Plan-Apochromat 63×/1.4 numerical aperture oil immersion objective (Carl Zeiss Inc, Jena, Germany) with the optical slice set to 1 μm. Images were processed by MetaMorph software (Molecular Devices, San Jose, CA). Line scan images at 2 Hz were recorded from ~22 mitochondria arrayed along individual myofibrils with excitation at 543 nm and collecting emission at >560 nm, and the confocal pinhole was set to obtain spatial resolutions of 0.4 μm in the horizontal plane and 1 μm in the axial dimension. Repetitive laser scanning of this row of mitochondria in a myocyte loaded with TMRM results in incremental, additive exposure of only the laser-exposed area to the photodynamic production of ROS and consequent mPTP induction. The occurrence of mPTP induction is clearly identified by the immediate dissipation of $\Delta\Psi$ in individual mitochondria and is seen at the point in time where "columns" of the line scan image suddenly lose TMRM fluorescence intensity and become black (*Figure 6*). The ROS threshold for mPTP induction (tmPTP) was determined as the average time necessary to induce mPTP in the exposed row mitochondria (N=3 in each genotype) (*Figure 6M and N*).

### Determination of autophagolysosome accumulation

Cardiomyocytes were loaded with the autophagy dye from CYTO-ID Autophagy detection kit (ENZO 51031-K200), according the manufacturer protocol (dilution 1:500 in HEPES buffer) and incubated for 30 min at room temperature. Then the dye was washed out by HEPES buffer and cell imaged with a confocal microscope (see above) in frame mode using 488 nm excitation and >505 nm emission filter. Images were processed by MetaMorph software. To discriminate the fluorescent spots representing autophagolysosomes from background and to ensure that all staining analyzed was only the true positive labelling, a defined threshold value was set for 488 nm excited fluorescence pixel intensity. The area of cell occupied by autophagolysosomes was expressed as fraction of total cell area (*Aon et al., 2021*).

### Source data

We provided source data for *Figures 2, 4–6*; *Figures 8, 12 and 13*. Proteome raw data submitted to MassIVE accession MSV000089554. RNASEQ raw data submitted to GEO GSE205234.

## Acknowledgements

We thank the NIDA Confocal and Electron Microscopy Core for access to their equipment, and specifically thank Zhang Shiliang (Core manager) and Marisela Morales (Director). We wish to thank Loretta Lakatta for her superb editorial assistance. Funding This research was supported by the Intramural Research Program of the NIH, National Institute on Aging (USA). LF is funded by NIH grant 1R01HL155218; NP by grants R01 HL136918 and R01 HL1155760; RGW by HL63030 and HL61912; MR by AHA grants #18CDA34110140 and #20TPA35500008.

## Additional information

### Funding

| Funder | Grant reference number | Author |
|---|---|---|
| National Heart, Lung, and Blood Institute | 1R01HL155218 | Loren Field |
| National Heart, Lung, and Blood Institute | R01 HL136918 | Nazareno Paolocci |
| National Heart, Lung, and Blood Institute | R01 HL1155760 | Nazareno Paolocci |
| National Heart, Lung, and Blood Institute | HL63030 | Robert Weiss |
| National Heart, Lung, and Blood Institute | HL61912 | Robert Weiss |

| Funder | Grant reference number | Author |
| --- | --- | --- |
| American Heart Association | #18CDA34110140 | Mark Ranek |
| American Heart Association | #20TPA35500008 | Mark Ranek |

The funders had no role in study design, data collection and interpretation, or the decision to submit the work for publication.

## Author contributions

Kirill V Tarasov, Conceptualization, Data curation, Formal analysis, Validation, Investigation, Visualization, Writing – original draft, Writing – review and editing; Khalid Chakir, Data curation, Formal analysis, Investigation, Visualization, Writing – review and editing; Daniel R Riordon, Formal analysis, Validation, Visualization, Writing – review and editing, Western Blot Analysis; Alexey E Lyashkov, Validation, Methodology, Writing – review and editing, Mass spectrometry Analysis; Ismayil Ahmet, Visualization, Methodology, Writing – review and editing, performed echocardiography and analysis; Maria Grazia Perino, Validation, Visualization, Writing – review and editing, Western Blot analyses; Allwin Jennifa Silvester, Validation, Visualization, Writing – review and editing, Western Blot Analyses; Jing Zhang, Validation, Visualization, Histological analysis; Mingyi Wang, Magdalena Juhaszova, Vikas Kumar, Mark Ranek, Dongmei Yang, Nazareno Paolocci, Validation, Visualization, Writing – review and editing; Yevgeniya O Lukyanenko, Validation, Visualization, Writing – review and editing, Sample preparation for Mass spectrometry Analysis; Adenylyl Cyclase Activity measurement; Jia-Hua Qu, Miguel Calvo-Rubio Barrera, Rostislav Bychkov, Visualization, Writing – review and editing; Yelena S Tarasova, Richard Telljohann, John Lammons, Gizem Keceli, Ashish Gupta, Walter Otu, Cameron Carroll, Validation, Visualization; Bruce Ziman, Validation, cardiac cells and tissues isolation; Rafael de Cabo, Miguel A Aon, Luigi Adamo, Steven J Sollott, Writing – review and editing; Seungho Jun, Thanh Huynh, Validation; Christopher H Morrell, Software, Visualization; Shane Chambers, Visualization; Karel Pacak, Robert Weiss, Loren Field, Validation, Writing – review and editing; Edward G Lakatta, Conceptualization, Resources, Supervision, Funding acquisition, Investigation, Methodology, Writing – original draft, Project administration, Writing – review and editing

## Author ORCIDs

Kirill V Tarasov http://orcid.org/0000-0001-7799-4670
Rafael de Cabo http://orcid.org/0000-0003-2830-5693
Gizem Keceli http://orcid.org/0000-0002-9562-7994
Edward G Lakatta http://orcid.org/0000-0002-4772-0035

## Ethics

All studies were performed in accordance with the Guide for the Care and Use of Laboratory Animals published by the National Institutes of Health (NIH Publication no. 85-23, revised 1996). The experimental protocols were approved by the Animal Care and Use Committee of the National Institutes of Health (protocol #441-LCS-2019).

## Decision letter and Author response

Decision letter https://doi.org/10.7554/eLife.80949.sa1
Author response https://doi.org/10.7554/eLife.80949.sa2

# Additional files

## Supplementary files

• Supplementary file 1. List of: Echo parameters, transcripts and proteins, canonical pathways, cardiovascular disease-related functions within the LV transcriptome and proteome of TG[AC8] and WT, RT-qPCR primers and a antibodies used in current study. (a) List of Echo parameters recorded in TG[AC8] heart vs WT. (b) A Listing of all 11810 identified transcripts. (c) A listing of 2323 transcripts that significantly differed in expression by genotype. (d) A listing of all 6834 identified proteins. (e) A listing of 2184 proteins that significantly differed in expression by genotype. (f) A listing of canonical pathways in IPA analysis of the total number of transcripts. (g) A listing of canonical pathways in IPA analysis of the total number of proteins that were differentially enriched or activated

in TG^{AC8} vs WT. (h) A listing of molecules with indication of involvement in number of pathways. (i) A listing of 544 molecules of which both transcripts and proteins differed by genotype. (j) A listing of canonical pathways that were differentially enriched by genotype in IPA analysis of 544 transcripts and proteins. (k) A Complete list of downstream effects of enriched canonical signaling pathways in TG^{AC8}, depicted on *Figure 10*. (l) RT-qPCR analysis of selected transcripts related to G-protein Coupled Receptor Signaling, that differed by genotype in RNASEQ. (m) Transcripts and proteins involved in Cell Cycle/Cell Proliferation, and Growth and Developmental Circuits that differed between TG^{AC8} vs WT. (n) IPA representation of top cardiovascular disease-related functions within the LV transcriptome of TG^{AC8} and WT. (o) IPA representation of top cardiovascular disease-related functions within the LV proteome of TG^{AC8} and WT. (p) Primers used in RT-qPCR analyses. (q) Antibodies used in WB analyses, antibody arrays and immunostaining.

- MDAR checklist

## Data availability

RNASEQ raw data have been deposited in GEO under accession code GSE205234. Proteome raw data have been submitted to MassIVE MSV000089554.

The following datasets were generated:

| Author(s) | Year | Dataset title | Dataset URL | Database and Identifier |
|---|---|---|---|---|
| Tarasov KV, Chakir K, Tarasova YS, Lakatta EG | 2022 | RNA-Seq Analyses of TGAC8 and WT mouse Left Ventricles | https://www.ncbi.nlm.nih.gov/geo/query/acc.cgi?acc=GSE205234 | NCBI Gene Expression Omnibus, GSE205234 |
| Tarasov KV | 2022 | Proteome Analyses of TGAC8 and WT mouse Left Ventricles | https://massive.ucsd.edu/ProteoSAFe/dataset.jsp?task=dee8806401274994945e09412adc4a8f | MassIVE, MSV000089554 |

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
