## [Editor Report]

The study is overall well-planned and the amount of data presented by the authors is impressive. The work nicely incorporates animal-level physiology (echocardiography data), tests for known canonical markers of hypertrophy, and then delves into an unbiased analysis of the transcriptome and proteome of LV tissue in bulk. The techniques and analyses in the study are adequately executed and within the realm of expertise of the Lakatta laboratory. This study is a necessary and crucial first step to extensively phenotype this mouse line and generate hypotheses for further work.

---

## [Decision Letter]

**Decision letter after peer review:**

Thank you for submitting your article "A Remarkable Adaptive Paradigm of Heart Performance and Protection Emerges in Response to the Constitutive Challenge of Marked Cardiac-Specific Overexpression of Adenylyl Cyclase Type 8" for consideration by *eLife*. Your article has been reviewed by 3 peer reviewers, and the evaluation has been overseen by a Reviewing Editor and a Senior Editor. The following individual involved in review of your submission has agreed to reveal their identity: Kalyanam Shivkumar (Reviewer #1).

The reviewers have discussed their reviews with one another, and the Reviewing Editor has drafted this letter to help you prepare a revised submission.

Essential revisions:

The authors should strive to shorten the paper a bit. The cancer data is an area where the paper can be shortened. The specific comments are given below and the reviewers have sought clarification of the data.

*Reviewer #1 (Recommendations for the authors):*

1. The present study provides an impressive breadth and depth of data characterizing the TGAC8 mouse line, with 16 primary figures. I believe the manuscript will be much greater impact if the authors can distill the core message and primary findings to better engage readers. The authors can consider removing certain figures since previously published (ie, Figure 5C) or moving parts to the data supplement (Figure 1, 13, 14, 15).

2. The connection between this study and cancer is less clear, which appears to be primarily based on the transcriptomic and proteomic studies. While malignancy and the cardiovascular phenotype in this mouse model share certain cellular properties (cell proliferation, inflammation, changes in metabolic profiles), the relationship between the two is hypothesis generating.

3. Do the authors have any single-cell sequencing data from the mouse LV? Of particular interest would be the interstitial non-cardiac myocytes that were identified in Figure 3. How do these cell types differ from cardiomyocytes? This may be an area of future direction as may not be feasible to incorporate additional data into the present study.

4. How did the authors select the mouse age and time frame for study in the present work? Have the authors compared these results to when TGAC8 develops decompensated heart failure?

*Reviewer #2 (Recommendations for the authors):*

1. Some of the data presented appear similar to that presented in the group's Moen et al. Fronters in Neuroscience 2019 publication. Could some of the data be referenced to that paper rather than re-presented here (e.g. Figure 5C has a direct citation to that paper)?

2. The manuscript reads quite long. In particular, the pathways covered in the discussion with numerous references to figures reads more like a Results section. Recommend focusing the discussion on the most salient points.

*Reviewer #3 (Recommendations for the authors):*

I enjoyed reading the manuscript and it certainly made me think!

General

1. Ultimately, this is a descriptive and hypothesis-generating study rather than providing directly proven mechanistic insights. There are some key questions that I asked myself (see above) which would as a minimum bear some discussion and would allow the wider significance of the findings to be assessed. Can the authors also discuss whether some or all of the changes they observe have been previously found in other transgenic models?

2. Can any of the pathways that are apparently activated be directly perturbed to assess whether they are mechanistically important?

Specific

1. A fundamental piece of information required is the strain of the mice and their sex please. Also, the exact age at which the various assays were performed.

2. How (or when) did this increase in cell number occur? Assessing the mice at birth (e.g. structure, cell number, perinatal proliferative window) might help.

3. Metabolism – ideally, any robust conclusions need to be based on an assessment of activity of key pathways (e.g. by metabolic flux) or perhaps metabolomics (less direct). As a minimum, this key limitation should be acknowledged and the conclusions moderated.

4. Apparently, the mice ultimately develop heart failure at around 12 months of age. Do the authors know what changes?

5. I had to educate myself and look up the meaning of consilience, which seems to be a favorite word! Based on the definition in Websters, are the authors confident that it has been used appropriately in every place?

---

## [Author Response]

Essential revisions:The authors should strive to shorten the paper a bit. The cancer data is an area where the paper can be shortened. The specific comments are given below and the reviewers have sought clarification of the data.Reviewer #1 (Recommendations for the authors):1. The present study provides an impressive breadth and depth of data characterizing the TGAC8 mouse line, with 16 primary figures. I believe the manuscript will be much greater impact if the authors can distill the core message and primary findings to better engage readers. The authors can consider removing certain figures since previously published (ie, Figure 5C) or moving parts to the data supplement (Figure 1, 13, 14, 15).

Thank you for this important message- we needed to hear it. We tried to distill the core message and primary findings, in the revised manuscript to better engage the readers. To this end, we have moved Figures 5, 12 and 13 to the supplement and in addition, we have deleted Figure 5C. We consolidated the Discussion section from 5716 words to 5188 words. Also, we added one sentence (Lanes 251-253), describing Figure 9 and incorporated Supplementary Methods into the main text/main methods. We have attached Word file with all changes in tracking mode.

2. The connection between this study and cancer is less clear, which appears to be primarily based on the transcriptomic and proteomic studies. While malignancy and the cardiovascular phenotype in this mouse model share certain cellular properties (cell proliferation, inflammation, changes in metabolic profiles), the relationship between the two is hypothesis generating.

We agree that the connection made between our results and cancer in the original manuscript was somewhat confusing and highlighting this connection may have appeared to hypothesis generating. We have removed reference to cancer from the Abstract and have referred to it only in one part of the Discussion, which we have rewritten for clarity, as follows:

When we attempted to ascertain the extent to which cardiac protection circuits identified in the TG^AC8^ LV (Figure 13) might be utilized as adaptations to the stress of various disease states, the top disease categories identified in both the TG^AC8^ transcriptome and proteome were Organismal Injury and Abnormalities and Cancer, the latter category being subsumed within the former. Because it is well known that cancers of the heart are rare, the implication of the patterns of “cancer” in the TG^AC8^ LV based on our bioinformatics analytic tools (IPA analyses) was at first puzzling to us, so we tried to rationalize what that might mean. But, as pointed out by Weinberg and Hanahan in their “Hallmarks of Cancer” ^15, 16^ mammalian cells have evolved complex concentric circuits to defend against stress that threatens their health or life, and envisioned that cancer cells successfully breach one or more of these concentrically arrayed defenses, because of somatic mutations in genes coding for proteins within this adaptive circuitry. They referred to this ability of cancer cells to breach this defense circuitry as the “Hallmarks of Cancer,” which, in reality, mirrored aspects of the mammalian complex defense circuitry that had been breached.

Thus, we envisioned that the overwhelming cardiac stress within heart cells of chronic intense AC/PKA/Ca^2+^ signaling, due to the expression of a single gene, AC8, engages this conserved complex mammalian concentric defense circuitry within heart cells that is embodied in over 250 canonical signaling pathways that were differentially enriched or activated in TG^AC8^ vs WT. Numerous processes within these pathways, e.g., protection from apoptosis, enhanced survival, engagement of aerobic glycolysis, increased proliferation, enhanced protein synthesis and quality control, increased inflammatory and immune response, enhanced tissue damage repair and regeneration because the sequalae of marked chronically increased AC/cAMP/PKA signaling illustrated in TG^AC8^ LV protection circuitry (Figure 13), are embodied within the “Hallmarks of Cancer”. ^15, 16^ We envision that engagement of this concentric protection circuitry in TG^AC8^ allows its heart to cope (for at least up to a year with numerous, marked cell-tissue stressors driven by the markedly augmented AC-cAMP-PKA-Ca^2+^ signaling).^10^ “Very limited versions” of this emergent cardio-protective profile in the TG^AC8^ LV have been previously discovered (by reductionist biologic research) to be central to cardiac ischemic pre-conditioning^88^ and exercise endurance cardiac conditioning.^1,89^

When our omics data reflecting engagement of concentric adaptive circuits within the TG^AC8^ LV but were specifically filtered for “cardiovascular system” and “cardiovascular disease,” bioinformatic analyses predicted features commonly associated with a variety of cardiomyopathies (Table S.11 A, B). The basis of this prediction, to us, seems likely to stem from the fact that omics “insights”, generated by findings in prior cross-sectional studies of experimental heart failure, may have often been misinterpreted to be causal factors of the heart failure, rather than adaptations that are activated to compensate for the causal factors of heart failure, due to a lack of longitudinal prospective.

Chronic consistent engagement of numerous adaptive pathways, however, may present a double-edged sword, compensating for the stress of chronic myocardial performance in the “short run”, but becoming maladaptive when utilized over prolonged periods of time. In fact, TG^AC8^ mice have a reduced (about 30%) ^10^ median life span, and later in life (about 12 mo) marked LV fibrosis and dilated cardiomyopathy become manifest. Thus studies, conducted longitudinally over the TG^AC8^ life course, are required in order to determine which of these correlated findings are cause-and-effect vs those that are simply associated. In other terms, longitudinal analysis are required to dissect out which mechanism within the “super performing heart” of the TG^AC8^ during young adulthood gradually fail over a prolonged period of time, resulting in accelerated cardiac aging and severe, dilated cardiomyopathy.^10^

We trust that revising the Discussion in this way clarifies what we mean by our reference to cancer in this context. Of note, the majority of studies in the cardiac literature these days, being somewhat reductionist in nature, usually focus on one or another pathway of this concentric stress adaptation circuitry. Here, we have been able to achieve a broad view of concentric adaptive circuity in the TGAC8 heart and feel that it is important to preserve this idea in our paper, because the relationship between the two is hypothesis generating and is a new way of thinking about the heart’s exquisite, complex, built-in, *concentric circuitry* to cope with stress. But if you still advise us to remove any reference to cancer in the manuscript, we, of course, will comply.

3. Do the authors have any single-cell sequencing data from the mouse LV? Of particular interest would be the interstitial non-cardiac myocytes that were identified in Figure 3. How do these cell types differ from cardiomyocytes? This may be an area of future direction as may not be feasible to incorporate additional data into the present study.

Presently, we do not have any single-cell sequencing data, but because our discovery on interstitial non-cardiac myocytes is extremely novel, we are about to launch single-cell/single-nuclear RNAseq of both the TGAC8 SAN and LV. We believe that interstitial non-cardiac myocytes population is part of a major “stem cell niche” in the heart, renewing non-cardiomyocytes that produce cytokines/chemokines that are essential to the preservation of myocardial health.

4. How did the authors select the mouse age and time frame for study in the present work? Have the authors compared these results to when TGAC8 develops decompensated heart failure?

We engaged in studies of the TGAC8 mouse because of our prior discovery that the “biochemical engine” driving spontaneous action potential firing in sinoatrial nodes is adenylyl cyclase type 8 (AC8). We chose 3 months of age because it was the approximate age of earlier studies of the Hanoune group (Circulation 2000) at INSERM in Créteil who created this mouse. We first sought to determine the role of AC8 in heart rate in vivo*,* and implanted these mice at 3 months of age with telemeters, and uncovered not only a marked increase in heart rate around the clock, but also a markedly coherent heart rate variability pattern (Moen reference). We went on to thoroughly characterize the cardiac LV structure and function phenotype via echocardiographic assessment of cardiac structure and function, and performed numerous assays of LV cells/tissue, including multi-omics that are embodied in the present manuscript.

Reviewer #2 (Recommendations for the authors):1. Some of the data presented appear similar to that presented in the group's Moen et al. Fronters in Neuroscience 2019 publication. Could some of the data be referenced to that paper rather than re-presented here (e.g. Figure 5C has a direct citation to that paper)?

The results in Figure 5A and B are novel and apply to the LV. You are correct that Figure 5C, ie. plasma levels of catecholamines and has been published previously in conjunction with our study on heart rate and heart rate variability in the AC8 mouse in vivo (Moen et al. Frontiers in Neuroscience 2019). We have removed Figure 5C from the manuscript.

2. The manuscript reads quite long. In particular, the pathways covered in the discussion with numerous references to figures reads more like a Results section. Recommend focusing the discussion on the most salient points.

Thank you for this important comment. In the discussion of the revised manuscript, we have deleted most of the references to Figures in the Discussion section, and we did our best to focus on the most salient points of our study. We consolidated the discussion from 5716 words to 5188 words and reorganized order of subheaders in Discussion section. We have attached Word file with all changes in tracking mode.

Reviewer #3 (Recommendations for the authors):I enjoyed reading the manuscript and it certainly made me think!

We are delighted that you enjoyed reading the manuscript. Many others that have read, including many of the authors, found it to be overbearing due to its broad scope. We are extremely delighted that the manuscript “made you think” as you put it. We have streamlined the revised version to make it more accessible to the reader.

General1. Ultimately, this is a descriptive and hypothesis-generating study rather than providing directly proven mechanistic insights. There are some key questions that I asked myself (see above) which would as a minimum bear some discussion and would allow the wider significance of the findings to be assessed. Can the authors also discuss whether some or all of the changes they observe have been previously found in other transgenic models?

In our opinion, AC8 mouse model is very relevant to stress adaptation in general, but this broad view has hardly ever been realized previously in the literature, because of the reductionist nature (by necessity) of mainstream biomedical research. For example, reports on cardiac specific overexpression of AC5 and AC6 never provided broader view on these mice and were focused only on a limited number of traits i.e., arrhythmogenesis, chronic pressure overload, contraction (Am J Physiol Heart Circ Physiol. 2015 Feb 1;308(3):H240-9; Am J Physiol Heart Circ Physiol. 2010 Sep;299(3):H707-12; Clin Transl Sci. 2008 Dec;1(3):221-7; Proc Natl Acad Sci U S A. 2003 Aug 19;100(17):9986-90; Am J Physiol Heart Circ Physiol. 2013 Jul 1;305(1):H1-8).

2. Can any of the pathways that are apparently activated be directly perturbed to assess whether they are mechanistically important?

Yes, many of the pathways that are activated in TGAC8 can be perturbed to assess whether they are mechanistically important. Our approach was initially to define the circuitry of numerous interconnected stress responses, as we did in the present manuscript. Following that, our goal is to perturb several select pathways to investigate their specific mechanisms in the TGAC8 stress response. We have already begun to do this in numerous pathways including proteostasis, autophagy, ROS defenses and metabolism. But these studies will require substantial time to complete. Another major goal of our lab is to define how the elements of the stress response circuitry fails as the TGAC8 mouse ages, in part, by replicating the systems approach and assays used in the present study in mice of different ages, followed by specific mechanistic studies. See response to Reviewer 2.

Specific1. A fundamental piece of information required is the strain of the mice and their sex please. Also, the exact age at which the various assays were performed.

All assays were performed on 3-month-old males. This information was inadvertently not directly stated in the original submission.

2. How (or when) did this increase in cell number occur? Assessing the mice at birth (e.g. structure, cell number, perinatal proliferative window) might help.

It is likely that an increase in number of cardiomyocytes may have occurred during the embryonic stage of development (8.5 dpc), when AC8 expression begins. Since submitting our manuscript we have found that the expression level of human AC8 (the type of AC8 employed in this transgenic model) increases markedly during the embryonic period when compared to endogenous AC8 and remains elevated in both the fetal and perinatal periods.

3. Metabolism – ideally, any robust conclusions need to be based on an assessment of activity of key pathways (e.g. by metabolic flux) or perhaps metabolomics (less direct). As a minimum, this key limitation should be acknowledged and the conclusions moderated.

We concur and have described some of the types of metabolic assessments in the last section of our discussion “Opportunities for Future Scientific Inquiry Afforded by the Present Results”: “… precisely defining shifts in metabolism within the cell types that comprise the TGAC8 LV myocardium via metabolomic analyses, including fluxomics.97 It will be also important that future metabolomics studies elucidate post-translational modifications (e.g. phosphorylation, acetylation, ubiquitination and 14-3-3 binding) of specific metabolic enzymes of the TGAC8 LV, and how these modifications affect their enzymatic activity”.

4. Apparently, the mice ultimately develop heart failure at around 12 months of age. Do the authors know what changes?5. I had to educate myself and look up the meaning of consilience, which seems to be a favorite word! Based on the definition in Websters, are the authors confident that it has been used appropriately in every place?

We are indeed using the Merriam-Webster definition of consilience as the linking together of principles from different disciplines especially when forming a comprehensive theory. We used the word consilience as a noun or consilient as an adjective numerous time in the manuscript. Our intent was to use the term to link together numerous pathways that concurrently become activated to protect the TGAC8 heart, i.e., a consilience of activated pathways. In the revised text, in numerous places, we have substituted consilience with the word concentric, e.g., mammalian cells have evolved complex concentric circuits to defend against stress that threatens their health or life.